J Physiol 604.1 (2026) pp 503–526

# Causal disconnectomics of motion perception networks: insights from transcranial magnetic stimulation-induced BOLD responses

Estelle Raffin[1,2,3] , Roberto F. Salamanca-Giron[1,2] , Krystel R. Huxlin[4] , Olivier Reynaud[5], Loan Mattera[5], Roberto Martuzzi[5] and Friedhelm C. Hummel[1,2,6]

[1] Neuro-X Institute (INX), Defitech Chair of Clinical Neuroengineering, Ecole Polytechnique Fédérale de Lausanne (EPFL), Geneva, Switzerland
[2] INX, Defitech Chair for Clinical Neuroengineering, EPFL Valais, Clinique Romande de Réadaptation, Sion, Switzerland
[3] LPNC, Université Grenoble Alpes, CNRS UMR 5105, Grenoble Cedex 09, France
[4] The Flaum Eye Institute and Center for Visual Science, University of Rochester, Rochester, New York, USA
[5] MRI Platform, Fondation Campus Biotech Geneva (FCBG), Campus Biotech, Geneva, Switzerland
[6] Department of Clinical Neurosciences, Geneva University Hospital (HUG), Geneva, Switzerland

Handling Editors: Richard Carson & Bettina Schwab

The peer review history is available in the Supporting Information section of this article (https://doi.org/10.1113/JP289699#support-information-section).

The Journal of Physiology

**Abstract figure legend** Differential behavioural and network effects of transcranial magnetic stimulation applied to the early visual areas compared to the motion-sensitive medio-temporal cortex. EVA, early visual areas; MT, medio-temporal cortex; TMS, transcranial magnetic stimulation.

---

This article was first published as a preprint. Raffin E, Salamanca-Giron RF, Huxlin KR, Reynaud O, Mattera L, Martuzzi R, Hummel FC. 2022. Causal disconnectomics of motion perception: insights from TMS-induced BOLD responses. bioRxiv. https://doi.org/10.1101/2022.03.03.482512

**Abstract** Understanding how focal perturbations trigger large-scale network reorganization is essential for uncovering the neural mechanisms that support perception and behaviour. Here we used a transcranial magnetic stimulation (TMS) perturbational approach by applying brief 10 Hz TMS to early visual areas (EVAs) or the medio-temporal (MT) area in healthy participants while recording concurrent functional magnetic resonance imaging (fMRI). TMS delivered during the early stages of motion processing specifically impaired direction discrimination at both sites, whereas disruption of the later processing phase impaired performances only for the MT condition. Despite a similar local increase in BOLD activity induced by EVA and MT stimulation, the broader network responses diverged significantly. Perturbation of EVA elicited a more robust and efficient pattern of functional reorganization, manifesting as more constrained BOLD changes, consistent with greater resilience to focal disruption. In contrast behavioural impairments induced by MT stimulation were accompanied by a disorganized and less-efficient network configuration, characterized by smaller small-world properties and longer path lengths. The decrease in performances induced by MT stimulation scaled with lower clustering coefficients, implying a more random or decentralized network structure. These findings demonstrate that TMS-fMRI coupling provides a powerful framework for causally mapping the relationships between local neural perturbations, large-scale network dynamics and behavioural performance.

(Received 11 July 2025; accepted after revision 4 November 2025; first published online 23 November 2025)

**Corresponding author** E. Raffin: LPNC, CNRS UMR 5105, Université Grenoble Alpes, Saint-Martin-d'Hères, France. Email: estelle.raffin@univ-grenoble-alpes.fr

## Key points

- Transcranial magnetic stimulation (TMS)-induced perturbation of the early visual areas (EVAs) or the medio-temporal (MT) area selectively impairs motion direction discrimination.
- The TMS perturbation is associated with a context-dependent local upscaling of blood-oxygen-level dependent (BOLD) activity in both areas.
- The two visual areas exhibit distinct topological networks' adaptation in response to TMS, reflecting different levels of network resilience to a focal lesion.
- TMS–functional magnetic resonance imaging (fMRI) coupling can be used to assess 'causal disconnectomics' and to precisely map how a local perturbation propagates to large-scale behavioural deficits.

## Introduction

Brain networks dynamically adjust their spatiotemporal organization to changing contexts and ongoing demands (Telesford et al., 2011; Papo et al., 2014). Such adaptive networks can be characterized by their topological transformation, their state dependency and/or their state transition. This adaptability of brain networks is crucial for maintaining an optimal balance between information segregation and integration, ensuring functional stability, efficient local processing and communication across local and long-range connections. A central challenge in systems neuroscience is to understand how flexible and distributed brain networks produce complex behaviour.

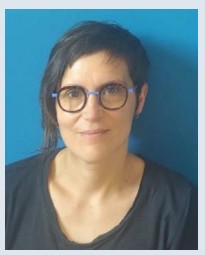

**Estelle Raffin** is a CNRS researcher at the University of Grenoble Alpes in France. She completed her PhD in cognitive neuroscience. Dr Raffin is particularly interested in exploring different forms of brain plasticity at the system's level, whether adaptive or maladaptive, that emerge after motor or perceptual learning, injury or neurological disease. The overarching goal of her work is to develop personalized, behavioural and neurotechnology-based interventions to promote functional recovery. To this end she employs innovative methodologies such as real-time coupling of non-invasive brain stimulation (magnetic or ultrasound) with brain imaging (functional magnetic resonance imaging (fMRI), EEG), together with advanced analyses of neural signals.

Although advances in neuroimaging have provided detailed descriptions of functional and structural brain connectivity (Lim et al., 2019; Ma et al., 2022; Litwińczuk et al., 2022; Blanco et al., 2024), these correlational approaches offer limited insights into the causal mechanisms that underlie network dynamics and behaviour, resulting in an incomplete account of how focal perturbations causally reshape large-scale network dynamics.

Addressing this gap requires experimental models in which neural activity can be perturbed in a highly controlled manner, allowing direct tests of how local disruptions propagate through distributed circuits. This study adopts such an approach, providing a proof-of-principle framework for investigating causal network reorganization using concurrent transcranial magnetic stimulation (TMS) with functional magnetic resonance imaging (fMRI) (Leitão et al., 2015; Vink et al., 2018; Bergmann et al., 2021). The combination of TMS with fMRI provides a unique opportunity to temporarily perturb a brain region and assess the changes in local brain activity and the resulting large-scale network responses *in vivo* that may underlie changes in task performance (Ruff et al., 2006; Bestmann S., 2008; Grosshagauer et al., 2024; Glick et al., 2025). Several attempts have been made to combine fMRI with TMS to capture the impact that TMS can have on patterns of functional connectivity in specialized, large-scale, brain systems (Ruff et al., 2008; Jung et al., 2020). This holds great promise for establishing *individual causal connectomics* (Glick et al., 2024), the ability to map the local effects of targeted perturbations of brain nodes and how they propagate across inter-connected networks in individual participants.

Despite its conceptual appeal key physiological mechanisms underlying TMS–fMRI responses remain incompletely understood. First the local neural response to TMS reflects a complex interplay between intrinsic cortical excitability, synaptic connectivity, ongoing oscillatory activity and network state. Second, the associated local TMS-evoked BOLD response and its physiological underpinnings remain to be fully elucidated (Rafiei & Rahnev, 2022). However earlier studies have shown that these local perturbations can propagate through polysynaptic pathways, producing remote BOLD responses that seem to reflect both direct anatomical projections and state-dependent functional interactions (Ruff et al., 2006; Bestmann S., 2008; Siebner et al., 2022). The distinct propagation patterns across brain regions, combined with individual variability, offer a valuable window into mapping subject-specific causal connectivity.

Describing personalized causal network mechanisms is particularly relevant when considering the application of TMS–fMRI to complex behavioural domains such as perceptual decision-making and motion direction discrimination (MDD). MDD is supported by a distributed network that integrates early visual areas (EVAs, V1/V2) (Moore et al., 2001; Koivisto et al., 2010), motion-sensitive temporal regions (medio-temporal (MT)) (Newsome & Paré, 1988; Liu & Pack, 2017), the intraparietal sulcus (IPS) involved in sensory evidence accumulation (Chen et al., 2017; Zhang et al., 2022; Wongtrakun et al., 2025) and prefrontal circuits supporting decision-making and response selection (Kennerley & Walton, 2011; Lin et al., 2020). Impairments in this system can result from dysfunctions at multiple network levels, leading to behavioural deficits that may be difficult to attribute to any single region. Disentangling whether such deficits arise from primary sensory dysfunction, impaired integration or altered top-down control requires tools that can interrogate how information flows across the network. By applying TMS–fMRI perturbations to key nodes within the motion discrimination network, here to the EVA and MT area, it becomes possible to experimentally probe how focal perturbations alter distributed network dynamics and behaviour. This approach may ultimately allow us to characterize individual-specific network vulnerabilities that shape multidomain interactions, offering insights into how sensory, cognitive and motor processes emerge from shared and distributed neural dynamics or even result in multidomain symptoms in clinical populations where sensory, cognitive and motor symptoms often co-occur (Siegel et al., 2016; Fleury et al., 2022; Jimenez-Marin et al., 2022). However achieving such mechanistic under-standing requires further efforts to relate region-specific TMS-evoked local and network-level responses to behavioural outcomes.

In this study we aim to address these challenges by combining an online TMS perturbational approach targeting EVA and MT with fMRI to more accurately address the causal and state-dependent mechanisms of perceptual decision-making network adaptation. These two regions were chosen because they allowed us to compare the whole-brain consequences of a TMS perturbation affecting a primary integrative brain region, acting as a gateway to other 'higher' visual areas and, in contrast, a more functionally specialized brain region (Simoncelli & Heeger, 1998; Grossberg et al., 1999; Rust et al., 2006; Mineault et al., 2012). Simulation studies suggest that inhibitory repetitive TMS (rTMS) targeting peripheral regions may exert more pronounced acute changes in network organization than stimulation of hub or integrative regions (Gollo et al., 2017). Therefore we hypothesized that, as a primary integrative hub, EVA would exhibit greater local and network-level homeo-stasis after TMS, thereby attenuating the propagation of perturbation effects and limiting both behavioural impairments and widespread network reconfiguration compared to MT (Das et al., 2012; Gollo et al., 2017; Tu

et al., 2021). Therefore we anticipate that differences in TMS-induced behavioural impairments across behavioural state and across processing stages will be explained by distinct patterns of TMS signal propagation – shaped by the hierarchical network level of the targeted area – rather than by variations in local BOLD responses to TMS.

## Methods

### Participants

In total 31 healthy subjects (17 males, mean age: 28.8 years, range: 19–39 years) were selected for the experiment. All participants provided written informed consent prior to the experiment, and none of them met the MRI or TMS exclusion criteria (Rossi et al., 2021). This study was approved by the local Swiss Ethics Committee (2017-01761) and performed in accordance with the *Declaration of Helsinki*.

### Experimental design and task procedure

Sixteen participants (9 males, mean age: 26.5 years, range: 19–32 years) performed two TMS–fMRI sessions (except one dropout for the second session). During the first session TMS was applied to the right EVA (TMS$_{(EVA)}$: mean($\pm$SD) MNI (Montreal Neurological Institute) co-ordinate: 8(5); –76(4); 9(6)) using phosphene perception when possible (in 9 of 16 participants) or the O2 position based on the 10–20 EEG system; during the second session TMS was applied to the functionally defined right MT (TMS$_{(MT)}$: mean MNI co-ordinates (SD): 46(4); –83(6); 11(5)) (Fig. 1*A* and *B*). We chose this study design to maximize our chances to optimally target the individual MT area through a dedicated MT localizer acquired during the first session. For the two online TMS–fMRI sessions, TMS was applied during a *MDD task* and at *Rest*. These sessions had the same content except the anatomical T1-MPRAGE sequence and the MT functional localizer sequence, which were performed only during the first session. A short offline session prior to the MT session was needed to locate the individual MT cluster using a neuronavigation system (Localite GmbH, Bonn, Germany).

An offline control experiment was performed to control for TMS non-specific effects on additional 15 participants (7 males, mean age: 28.9 years, range: 22–39 years). We repeated the *MDD task* with TMS using the same set-up (neuronavigation and eye-tracking systems) and the same task parameters (see later) but stimulating the right motor cortex (TMS$_{(M1)}$, mean(SD) MNI co-ordinates: 53(7); –4(2); 54(3), corresponding to the individual first dorsal interossei muscle's motor hot spot of the participants),

outside the MRI scanner. Figure 1*C* summarizes the study timeline.

For the three sessions the *MDD task* consisted in rating subjective perceptual experience of visually presented motion stimuli and in performing a forced-choice direction discrimination task on the same moving stimuli (Koivisto et al., 2021). The visual stimuli involved a group of static or moving dots coherently to the right or to the left. During the familiarization phase outside the scanner, the duration of stimulus presentation was individually defined (either 50, 67 or 84 ms, corresponding to 3, 4, or 5 frames) to ensure ∼70% accuracy in the *noTMS* trials. The stimulus duration was then kept constant throughout the experiment (see Fig. 1*D* for an illustration of the task).

Just after the stimulus presentation participants were asked to rate their perception using a response box with their right hand using a 1–4 scale (see the caption of Figure 1 for more details). In the main experimental conditions TMS bursts were given to EVA when visual signals first reach the target region 60 ms after the onset of the visual stimulus (TMS$_{(EVA)}$) (Lamme, 2001) and 30 ms after stimulus onset (TMS$_{(MT)}$) for MT to account for the direct thalamic-extrastriate pathway (Beckers & Hömberg, 1992; Lamme & Roelfsema, 2000; d'Alfonso et al., 2002; Koivisto et al., 2010). This ensured similar perturbation of the feedforward thalamic inputs to the two regions. Later TMS onsets were also tested in half of the trials, 100 ms for EVA and 130 ms for MT selectively interfering with late processing stages. The total duration of the TMS bursts was 200 ms to cover the full window of visual processing in the two regions. TMS intensity was set to ≈80% (range: 75%–90%) maximal stimulator output (MSO), corresponding to a d$I$/d$t$ value of 119–169 A/μs. The resulting E-field magnitude at hot spot (99.9th percentile) corresponded to values between 73 and 112 V/m. This intensity was individually adjusted prior to the measurement to ensure phosphene sub-threshold stimulation and progressively increased until participants report pain or discomfort. The intensity was then chosen to obtain a reliable BOLD signal while preserving participants' comfort. For the M1 control experiment the same EVA timings were used, and TMS was given subthreshold to avoid motor evoked potential elicitation.

Conditions randomly alternated between TMS and noTMS trials, moving and static trials throughout the task, 50 trials each, giving a total of 300 trials. An inter-trial interval (ITI) of ∼1.5 s was used. The task was displayed in the scanner through a 44 × 27 cm LCD monitor at 2.5 m distance via a mirror mounted on the head coil on a frame on top of the TMS–fMRI set-up (Fig. 1*B*). For the offline M1 control experiment, the task was displayed on a mid-grey background LCD projector (1024 × 768, 144 Hz). To exclude bad performance due to TMS-induced blinking and to make sure that

participants were constantly looking at the fixation point, gaze and pupils' movements were controlled in real time using an EyeLink 1000 Plus Eye Tracking System (SR Research Ltd, Ottawa, Canada) in the scanner and in the behavioural room, sampling at a frequency of 1000 Hz. Trials were aborted and redone if deviation exceeded 1° from the fixation point. The duration of the *MDD Task-TMS* sequence was on average 16.5 min (SD = 4.30 min) depending on individual reaction times.

A *Rest-TMS* sequence was used to study the effect of TMS intensity. An event-related design was used to map the effect of TMS bursts composed of three pulses at $\alpha$ (10 Hz) frequency. Three conditions were pseudorandomized and counterbalanced across the run: high-intensity TMS (*HighTMS*), low-intensity TMS (*LowTMS*) and no TMS (*noTMS*), with 25 repetitions of each condition with an ITI of 6 s (covering 3 repetition times). *HighTMS* intensity corresponded to the intensity used during the MDD task, whereas *LowTMS* was set to ≈38% (35%–43%)

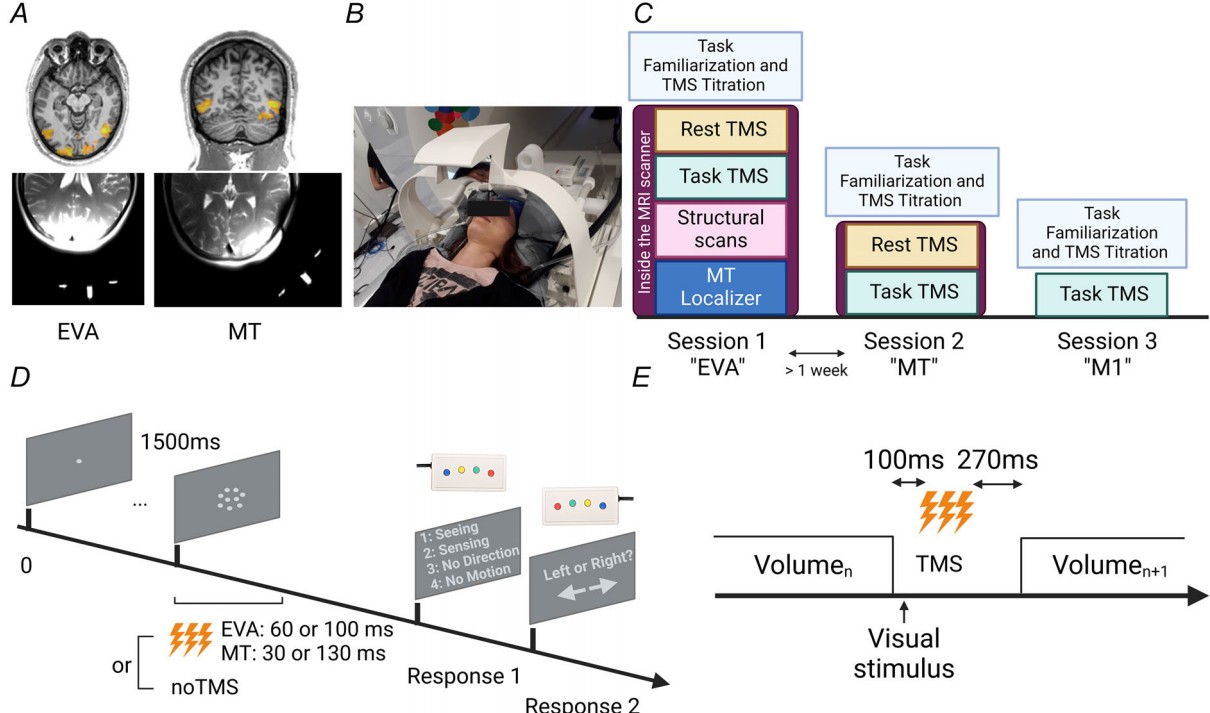

**Figure 1. Experimental set-up and design**

*A*, TMS (transcranial magnetic stimulation) targeting. TMS targets were determined by the functional localizer of EVA (early visual area) and MT (medio-temporal) for precise TMS positioning (see online materials for further details of the motion processing localizer). The two images on the right part show the T2 image of a representative subject with the oil capsules placed on the TMS-MRI (magnetic resonance imaging) coil casing to monitor the coil position located on the TMS coil for the right EVA and the right MT. These markers formed a 'T-like shape' indicating the centre of the coil as well as the coil orientation as seen on a T2 HASTE image obtained using the two TMS-compatible MRI coil arrays (56 axial slices, slice thickness = 0.8 mm, TR = 1500 ms, TE = 135 ms, FOV = 216 mm, flip angle = 140°). Coil positions were readjusted if needed. *B*, MRI set-up. The MRI-TMS coil was positioned over the occipital area, whereas the MR-receiving coil was positioned over the frontal lobe. Both were maintained via a vacuum pillow to ensure the absence of movement or vibration and to ensure a comfortable position for the participant. *C*, study timeline: description of the two experimental sessions; *D*, one single trial of the motion discrimination task. The visual stimuli involved a group of white dots in a 5° diameter circular aperture centred in the middle of the screen. The dots were either static or moving coherently to the right or to the left at a density of 2.6 dots per degree. Dots were displayed for 50, 67 or 84 ms (corresponding to 3, 4 or 5 frames). This duration was individually adjusted prior to the fMRI (functional MRI) measurement to ensure an ~70% accuracy in the noTMS trials. Just after the stimulus presentation participants were asked to subjectively rate their perception using a response box on their right hand with a 1–4 scale, 1 being 'I clearly saw the motion' (seeing), 2 being 'I sensed the motion' (sensing), 3 being 'I saw something moving but I'm unable to judge the direction' (no direction) and 4 being 'I didn't see any motion' (no motion) (Koivisto et al., 2010; Grasso et al., 2018). When the participants pressed the first 3 options, they were then asked to judge whether the dots moved to the right or to the left with a response box on their left hand. If participants pressed 4 the task moved to the next trial. *E*, schematic of the BOLD sequence allowing artefact-free combination of $\alpha$ TMS bursts.

MSO. Participants were asked to look at a fixation cross throughout the acquisition. The duration of the *Rest TMS* sequence was 9 min. Note that the results of the Rest-TMS sequence are not presented in this study but available in Zenodo at DOI 10.5281/zenodo.17175649.

## Behavioural data analysis

Considering static stimuli the mean motion perception accuracy, reflecting how well participants could distinguish between moving and static stimuli, was extracted as the percentage of correct answer relative to the total number of trials belonging to the same condition. Using moving stimuli MDD performances were extracted as the percentage of correct left–right judgements relative to the total number of trials of that condition. For both scores mixed ANOVA were built with factors Site (EVA, MT and M1) and TMS (Early onset, Late onset and NoTMS). Violation of sphericity was corrected using Greenhouse–Geisser corrections, and Tukey's multiple comparison tests were applied for *post hoc* analyses when significant main effects or interactions were found.

## MRI and TMS

MRI images were obtained at the MRI facility of the Human Neuroscience Platform of the Fondation Campus Biotech Genève (FCBG, Biotech Campus, Geneva, Switzerland) using a Siemens Prisma 3 Tesla scanner (Siemens Healthineers, Erlangen, Germany).

**Standard MRI sequence acquisition.** Anatomical images were obtained using a 64-channel head and neck coil using a 3-D MPRAGE sequence (TR/TE = 2300/2.96 ms, flip angle = 9°, 192 sagittal slices, matrix = 256, $1 \times 1 \times 1$ mm$^3$ resolution) covering the whole head. For retrospective distortion correction static field mapping was performed using a double-echo spoiled gradient echo sequence (TR = 652 ms, TE1/TE2 = 4.92/7.38 ms, resolution = $2.2 \times 2.2 \times 2.2$ mm$^3$, whole head coverage), generating two magnitude images and one image representing the phase difference between the two echoes.

A standard MT localizer task was acquired during the first fMRI session and used to accurately and individually target the MT area. The screen exhibited radially moving dots alternating with stationary dots (see, e.g. Sack et al., 2007). A block design alternated six 15-s blocks of radial motion with six blocks featuring stationary white dots in a circular region on a black background. This region subtended a visual angle of 25°, with 0.5 dots per square degree. Each dot was 0.36° diameter. In the motion condition the dots repeatedly moved radially inward for 2.5 s and outward for 2.5 s, with 100% coherence, at 20° per second measured at 15° from the centre. Participants were passively looking at the screen and were asked to focus on a fixation point located in the middle of the screen. The motion processing localizer was acquired using a grandient-echo (GE)-EPI sequence with 56 axial slices, slice thickness = 2.2 mm, in-plane resolution = 2.2 mm, TR = 2000 ms, TE = 30 ms, FOV = 242 mm, flip angle = 60°, GRAPPA = 2 and multiband (MB) factor = 2.

**Combined TMS–fMRI sequence acquisition.** For combined TMS–fMRI images two dedicated coil arrays were used (Navarro de Lara et al., 2017). This set-up consisted of an ultra-slim seven-channel receive-only coil array, which was placed between the subject's head and the TMS coil (MRI-B91, MagVenture, Farum, Denmark) and connected to a MagPro XP stimulator (MagVenture). A second, receive-only MR coil was positioned over Cz in the EEG 10–20 system to allow a full coverage of the participant's brain.

The *Rest-TMS* and MDD *Task-TMS* sequences were acquired with a GE-EPI sequence using the same parameters: 40 axial slices, slice thickness = 2.2 mm, in-plane resolution = 2.2 mm, TR = 2000 ms, TE = 30 ms, FOV = 242 mm, flip angle = 67°, GRAPPA = 2 and MB factor = 2. A gap was introduced between consecutive EPI volumes to ensure artefact-free MR images after TMS stimulation (Navarro de Lara et al., 2015). For both rest and task TMS sequences, a single repetition time (TR = 2000 ms) was therefore composed of 40 slices acquired during 1430 ms followed by a gap of 570 ms before the next volume acquisition (Fig. 1*E*). The TMS pulse was synchronized with an in-house script using MATLAB (R2019).

Static field mapping was also performed with the TMS-MRI coils using the same double-echo spoiled gradient echo sequence (TR = 652 ms, TE = 4.92 and 7.38 ms, slice thickness = 2.2 mm, in-plane resolution = 2.2 mm, flip angle = 60°) that generates two magnitude images and one image representing the phase difference between the two echoes. Functional and structural MR data, respectively, obtained using the TMS-MRI coils and the 64-channel head coil, were found difficult to co-register due to different coil coverage, sensitivity and contrast. Therefore two additional rapid high signal-to-noise, balanced, steady-state free pre-cession (SSFP) sequences (Bieri & Scheffler, 2013) (96 axial slices, slice thickness = 2 mm (gap = 0.4 mm), in-plane resolution = 2 mm, TR = 5.24 ms, FOV = 256 mm, flip angle = 28°) were obtained using both the TMS-MRI coils and the body coil integrated into the scanner bore to resolve, based on the same SSFP contrast, the coil coverage mismatch within the registration processing pipeline.

**Preprocessing of fMRI images.** Statistical Parametric Mapping software (SPM12, Wellcome Department of Imaging Neuroscience, London, UK) was used for data preprocessing and analysis of the four datasets (*Rest TMS* and *Task TMS* datasets, on both EVA and MT). Spatial distortions due to field inhomogeneities were removed from all EPI images using the FieldMap toolbox from SPM12 (Andersson et al., 2001). Subsequently all time series were motion corrected to the first volume using six degrees of freedom (3 rotations, 3 translations), and drifts in signal time courses were corrected as well. To improve the normalization procedure, two inter-mediate steps were added (see online materials for further details of the additional sequences used for co-registration purposes). A first co-registration was performed between the mean realigned and slice timing-corrected image and the SSFP sequence obtained using the same MR coil (and thus the same spatial coverage). The resulting, co-registered image was once more co-registered to the SSFP sequence obtained using the MR coil integrated into the scanner (i.e. the body coil, thus preserving the contrast). The latter could then be easily co-registered to the high-resolution T1 image obtained using the 64-channel head coil that covered the whole brain and later transformed into standard MNI space using a segmentation-based normalization approach (Ashburner & Friston, 2005). Finally the normalized images were spatially smoothed using a Gaussian kernel (4 mm, full-width half-maximal). The realignment parameters estimated during spatial preprocessing for the *Rest TMS* and *Task TMS* datasets were introduced in the design matrix as regressors of no interest to prevent confounding activations related to minor head movements during scanning. The experimental event-related designs were convolved with the canonical haemodynamic response function, and the resulting models described in the following section were estimated using a high-pass filter at 128 s to remove low-frequency artefacts.

**fMRI univariate analysis.** EVA and MT datasets were similarly analysed using general linear models (GLM) to calculate individual contrasts. For the *Task TMS* sequence a design matrix with TMS conditions (TMS$_{early}$, TMS$_{late}$, *noTMS*) and visual state (Moving, Static) was constructed using all trials. To disentangle the effects of TMS onsets and visual stimuli, we built GLM with random effects at the second level, by calculating a flexible factorial design, as implemented in SPM12, with the within-subject factors *TMS condition* (TMS$_{early}$, TMS$_{late}$) and Visual stimulus (Moving and Static).

To explore the local effects of TMS across conditions, mean $\beta$ values were extracted from the stimulated regions EVA and MT defined as a 8 mm-radius sphere based on the results of the group-level analysis of the motion processing localizer using the MarsBaR toolbox (Brett et al., 2002) (right EVA: 6, –83, 11 and right MT: 58, –66, –3). Repeated-measures ANOVAs were built with TMS (TMS$_{early}$, TMS$_{late}$, *noTMS*) and visual stimulus (Moving, Static) as within-subject factors for each site of stimulation (EVA and MT).

For the *RestTMS* sequence we defined a design matrix comprising three conditions (*HighTMS*, *LowTMS* and *noTMS*). A second-level GLM with random effects was designed using a flexible factorial design, with the within-subject factors *TMS intensity* (TMS$_{High}$, TMS$_{Low}$ and *noTMS*). $\beta$ values were extracted locally for each condition using the same ROI as previously described.

The statistical significance threshold was set to a height threshold of $P < 0.001$ uncorrected at the voxel level and to that of $P < 0.05$ at the cluster level after false-discovery rate (FDR) correction.

**fMRI multivariate analysis.** Independent component analysis (ICA) was used on the motion discrimination task data to estimate independent spatiotemporal functional networks from the data and to visualize networks specifically modulated by TMS. ICA uses fluctuations in BOLD signal to split it into maximally independent spatial maps or components, each explaining a unique part of variance from the four-dimensional fMRI data. Then all components have their specific time course related to a coherent neural signal potentially associated with intrinsic brain networks, artefacts or both. The group ICA of fMRI toolbox (GIFT version 3.0b; Calhoun et al., 2001) was used. First it concatenates the individual data and then computes subject-specific components and time course. Maximum description length (MDL) and Akaike's criteria were applied to estimate the number of ICs in our data. We then performed a subject-level principal component analysis (PCA) with the number of principal components as 16 on each subject's fMRI data and group-level PCA with the number of principal components as 8 on the reduced and concatenated data.

MDL and Akaike's criteria were applied to estimate the number of ICs in our data. Using PCA individual data were reduced. Then the infomax algorithm (Bell & Sejnowski, 1995) was applied on the PCA-reduced data for the group ICA and estimated eight components. This number is probably explained by the nature of online TMS–fMRI data which induces strong and stereotyped activity patterns that dominate the variance structure of the data, resulting in a lower effective dimensionality compared to resting-state acquisitions. To improve the IC's stability the ICASSO was applied and run 20 times (Himberg et al., 2004). Over the eight extracted network components, we focused on the TMS-specific networks defined by the overlap between the activated clusters and the co-ordinates of the stimulation target. Based on

these independent components (ICs), we then examined how each condition (Moving or Static) covaries with the TMS networks using the *temporal sorting* option in the GIFT toolbox. Temporal sorting consists in regressing the fMRI-specific time course for each individual against the design matrix for the experimental conditions. The resulting $\beta$ weights represent the degree to which the network was recruited by the conditions. For a network positive and negative $\beta$ weights indicate the level of network recruitment in each experimental condition. Paired-sample $t$ tests were used to identify networks that were significantly modulated by TMS at rest and during motion discrimination using $P < 0.05$.

**Brain network generation and graph theoretical analysis.** In addition to ICA we used the graph-theoretical network analysis toolbox GRETNA to image connectomics and construct the functional brain networks derived from the full Task $TMS_{(EVA)}$ and $TMS_{(MT)}$ datasets comprising all the conditions of interest (e.g. noTMS, TMS, motion or static stimuli; Wang et al., 2015). The whole brain was divided into 90 cortical and subcortical regions according to automated anatomical labelling (Tzourio-Mazoyer et al., 2002), and the mean time series for each of the 90 regions was extracted. Pearson's correlation coefficients for each pair of regions were calculated for the mean time series of all of the 90 regions and z-transformed using Fisher's Z transformation. Subsequently a positive binary undirected connection functional network was constructed according to a range of selected threshold values of the relation matrix. As there is no defined standard for threshold selection in the construction of a binary connection brain functional network, sparsity was used as a range of correlation coefficient thresholds for correlation metrics. It was defined as the existing number of edges in a graph divided by the maximum possible number of edges. In accord with previous studies we set both the sparsity and Pearson's correlation threshold of the functional network to range from 0.05 to 0.5 (in steps of 0.05), resulting in a more efficient functional network than a random network with the number of artificial edges minimized (Achard & Bullmore, 2007; Zhao et al., 2017). Graph-theoretical analysis was applied to assess the topological and organizational properties focusing on three graph-theoretical metrics (Rubinov & Sporns, 2010). These metrics were the clustering coefficient ($\gamma$), which measures how much neighbours of a node are connected to each other and is closely related to local efficiency; characteristic path length ($\lambda$), which is the average number of edges needed to get from any node in the network to any other node in the network and is inversely related to global efficiency; and small-worldness ($\delta$), which reflects whether the network balances global integration and local segregation for efficient information processing (Latora & Marchiori, 2001). To compare graph-theoretical network properties between EVA and MT, integration of the topological metric over all selected ranges of sparsity values was calculated using GRETNA. Paired $t$ tests corrected for FDR ($P < 0.05$) were performed between the topological measures of EVA and MT networks.

## Results

### Behavioural results

We used chronometric TMS–fMRI coupling in healthy participants to perturb and map the effect of TMS over the right EVA ($TMS_{(EVA)}$) and over the right MT ($TMS_{(MT)}$) during a MDD task. An additional offline experiment targeted the right primary motor cortex ($TMS_{(M1)}$) in another subgroup of healthy participants.

Figure 2*A* shows the group and individual motion perception accuracy, reflecting how well participants could distinguish between moving and static stimuli when a TMS burst was applied over EVA (second panel), MT (third panel) and the control region M1 (fourth panel). A global mixed ANOVA revealed no main effect of TMS ($F_{(1,24)} = 2.9$, $P = 0.084$, $\eta^2 = 0.02$, $n = 31$) and no significant TMS by Site interaction ($F_{(4,48)} = 0.9$, $P = 0.452$, $\eta^2 = 0.031$), suggesting that TMS has no effect on motion perception in these three regions. However there was a Site effect ($F_{(2,24)} = 6.8$, $P = 0.004$, $\eta^2 = 0.16$), likely driven by the overall better performances in the M1 session.

The same model applied to the MDD performances (Fig. 2*B*) revealed a significant TMS effect ($F_{(2,305)} = 22.012$, $P < 0.001$, $\eta^2 = 0.2$) and Site effect ($F_{(2,30)} = 20.57$, $P = 0.001$, $\eta^2 = 0.2$). There was also a TMS × Site interaction ($F_{(4,60)} = 9.87$, $P < 0.001$, $\eta^2 = 0.124$). When TMS was applied to EVA, performances were significantly affected by $TMS_{early}$ compared to noTMS ($t_{(15)} = 5.2$, adjusted $P < 0.001$) and compared to $TMS_{late}$ ($t_{(15)} = 4.4$, adjusted $P = 0.034$). Although slightly lower the difference between noTMS and $TMS_{late}$ was not significant ($t_{(15)} = 1.45$, $P = 0.35$). In contrast when applied to MT, $TMS_{early}$ and $TMS_{late}$ negatively impacted performance compared to noTMS ($TMS_{early}$: $t_{(15)} = 5.2$, adjusted $P < 0.001$, $TMS_{late}$: $t_{(15)} = 5.2$, $P < 0.001$). After correction for multiple comparisons performances under $TMS_{early}$ and $TMS_{late}$ were not different ($t_{(15)} = 2.2$, $P = 0.1$). Performances were similar in the three TMS conditions for M1 (noTMS *vs.* TMSearly: $t_{(14)} = 0.61$, adjusted $P = 0.55$; $TMS_{early}$ *vs.* $TMS_{late}$: $t_{(14)} = -1.5$, adjusted $P = 0.16$, noTMS *vs.* $TMS_{late}$: $t_{(14)} = -0.73$, $P = 0.475$).

Importantly baseline performances (noTMS conditions) were not different across sites ($TMS_{EVA}$ *vs.* $TMS_{MT}$: $t_{(15)} = 0.48$, $P = 0.64$; $TMS_{EVA}$ *vs.* $TMS_{M1}$:

$t_{(15)} = -0.28$, $P = 0.79$; $TMS_{MT}$ *vs.* $TMS_{M1}$: $t_{(15)} = -0.07$, $P = 0.9$).

In summary TMS bursts exerted negative effects on MDD at the early and late onsets in MT. TMS bursts were efficient only in disturbing MDD in the early onset for EVA. The two onsets did not affect performances when the control region M1 was targeted.

### Whole-brain and local responses to TMS perturbation

Next we analysed local BOLD activity in response to TMS bursts over EVA and MT applied during behavioural perturbation of MDD. First Fig. 3*A* ($TMS_{(EVA)}$) and Fig. 3*B* ($TMS_{(MT)}$) show the results of the average effects of TMS conditions. $TMS_{(EVA)}$ induced significant clusters in the bilateral occipital cortex, the right superior and middle temporal gyrus, the bilateral anterior and middle cingulate cortex, the left middle frontal gyrus, the bilateral caudate nucleus and the right posterior thalamus. When TMS was applied over the right MT, the average effect of TMS conditions returned significant clusters in the right occipital cortex, the right inferior and frontal gyrus, the right IPS and the right cerebellum (cruz VI). These clusters overlapped with the TMS-induced activation at rest.

We then examined the main effect of visual stimuli (difference between TMS applied during a moving or static stimuli) that returned significant clusters in the right middle temporal gyrus and in the bilateral middle frontal gyrus, the right superior frontal gyrus, the post-

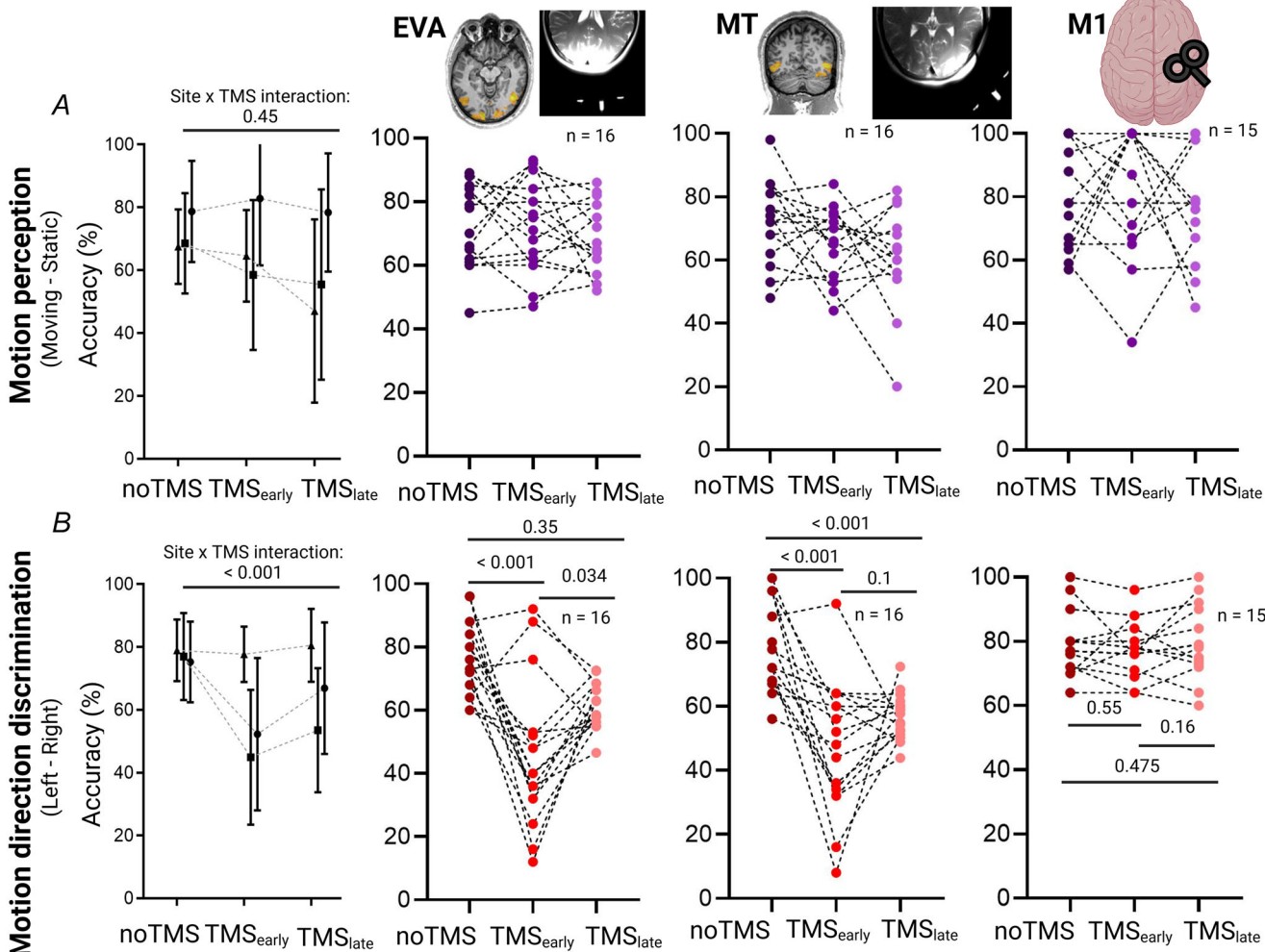

**Figure 2. TMS (transcranial magnetic stimulation)-induced behavioural perturbation**
*A*, group data (first column) and individual data of motion perception under noTMS, TMS_early and TMS_late for TMS_(EVA) (*n* = 16) (second column), TMS_(MT) (*n* = 16) (third column) and TMS_(M1) (*n* = 15) (fourth column) (non-significant TMS × Site interaction on a mixed ANOVA, *P* = 0.45); *B*, group data (first column) and individual data of motion direction discrimination under noTMS, TMS_early and TMS_late for TMS_(EVA) (*n* = 16) (second column), TMS_(MT) (*n* = 16) (third column) and TMS_(M1) (*n* = 15) (fourth column) (significant TMS × Site interaction on a mixed ANOVA, *P* < 0.001 and the associated *post hoc t* tests on each site).

erior cingulate cortex and the right occipital cortex when EVA was stimulated (Fig. 3*C*) and only in the right middle temporal gyrus when MT was stimulated (Fig. 3*D*). When the EVA was stimulated the main effect of TMS onsets revealed a significant cluster in the right middle frontal gyrus and in the right superior frontal gyrus (Fig. 3*E*). There was no significant cluster when MT was stimulated (see Table 1 for the MNI co-ordinates of the significant clusters of all effects).

The extracted $\beta$ values from the stimulated EVA and MT regions were analysed using an ANOVA test with the factors TMS (noTMS, TMS$_{early}$ and TMS$_{late}$),

Visual stimuli (moving, static) and stimulation site (EVA, MT). This analysis revealed a significant TMS effect ($F_{(2,22)} = 3.8, P = 0.038$), mostly driven by an increase in $\beta$ weights for the two TMS conditions compared to no TMS, whereas there was no difference between the early and late TMS onsets. Finally there was a significant TMS × Visual stimuli interaction ($F_{(2,22)} = 6.1, 0.003$), indicating that the increase in activity occurred only in the combination early TMS associated with moving stimuli (noTMS, Moving *vs.* early TMS, Moving: $t_{(12)} = -4.2, P = 0.003$, and early TMS, Static *vs.* early TMS, Moving: $t_{(12)} = -3.6, P = 0.015$; see Fig. 3*G* and Table 2 for all *post hoc* comparisons). There

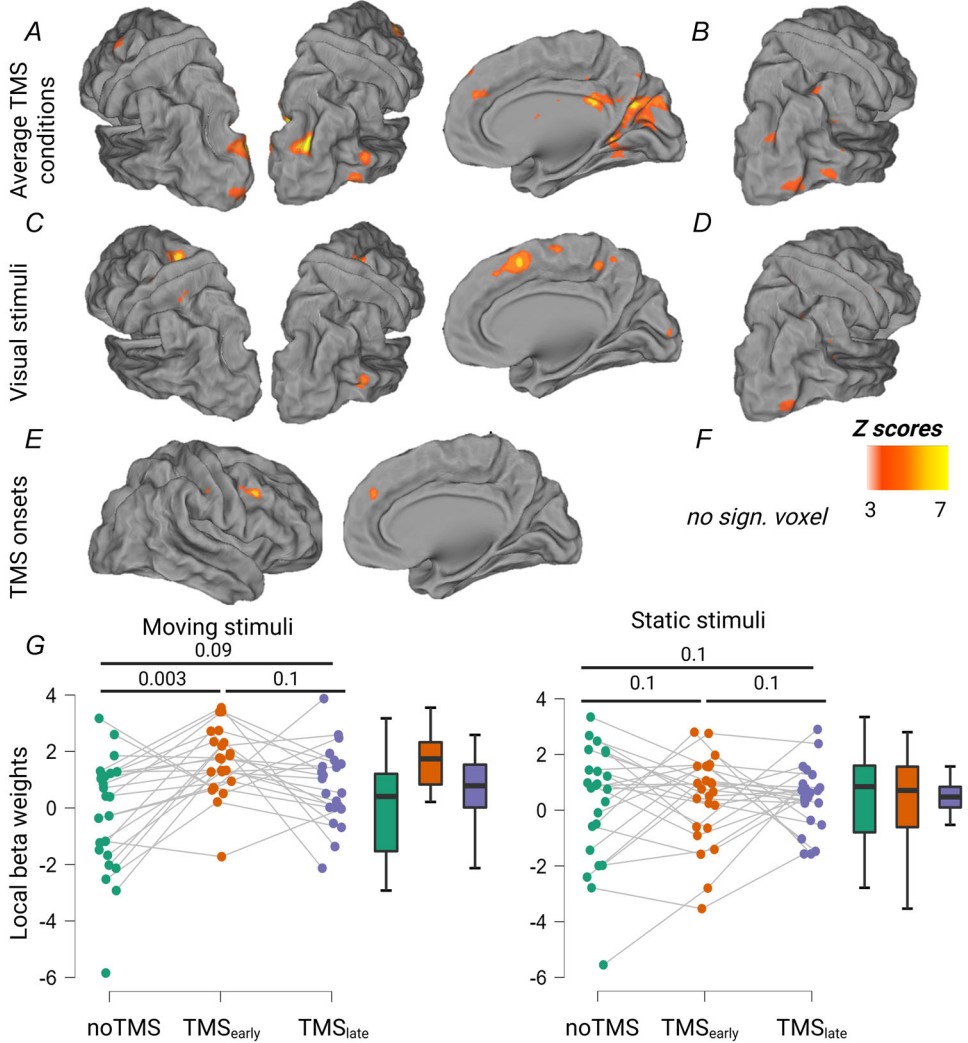

**Figure 3. TMS (transcranial magnetic stimulation)-induced activation maps**
*A*, activation maps corresponding to the average TMS conditions for TMS$_{(EVA)}$; *B*, activation maps corresponding to the average TMS conditions for TMS$_{(MT)}$; *C*, main effect of visual stimuli for TMS$_{(EVA)}$; *D*, main effect of visual stimuli for TMS$_{(MT)}$; *E*, main effect of TMS onsets for TMS$_{(EVA)}$; *F*, main effect of TMS onsets for TMS$_{(MT)}$; *G*, local $\beta$ weights extracted from the EVA (early visual area) and MT (medio-temporal) ROI during moving stimuli (left panel) and static stimuli (right panel) (*post hoc t* tests after a significant TMS × Visual stimuli interaction, $P = 0.008$, $n = 16$), $P$(Family-wise error (FWE)) < 0.001 at the cluster level. The colour bar indicates the group-level $Z$ scores.

**Table 1. Significant clusters evoked by TMS$_{(EVA)}$ and TMS$_{(MT)}$.**

| Region label | Extent | *F*-value | MNI co-ordinates | | |
| --- | --- | --- | --- | --- | --- |
| | | | *x* | *y* | *z* |
| *TMS(EVA)* | | | | | |
| *Visual stimulus effect* | | | | | |
| Right medial temporal cortex | 157 | 17.9797 | 46 | −52 | 0 |
| Right superior temporal gyrus | 137 | 15.2246 | 44 | −42 | 18 |
| Left middle frontal gyrus | 227 | 16.5663 | −20 | −12 | 60 |
| Right middle frontal gyrus | 433 | 13.0977 | 24 | −5 | 58 |
| Right precuneus | 311 | 15.5128 | 10 | −60 | 52 |
| Left precuneus | 311 | 15.1918 | −14 | −46 | 64 |
| Right posterior-medial frontal | 433 | 15.121 | 8 | 12 | 54 |
| *TMS(MT)* | | | | | |
| *TMS × Visual stimulus interaction* | | | | | |
| Left calcarine gyrus | 212 | 14.7565 | −10 | −84 | 14 |
| Right calcarine gyrus | 250 | 14.4692 | 10 | −86 | 8 |
| Right anterior cinculate cortex | 226 | 14.1312 | 4 | 44 | 34 |
| *TMS effect* | | | | | |
| Right IFG (pars triangularis) | 71 | 17.4877 | 32 | 14 | 32 |
| Right middle temporal cortex | 400 | 16.419 | 44 | −71 | −9 |
| Right precuneus | 363 | 15.4239 | 20 | −76 | 46 |
| Left caudate nucleus | 53 | 15.7919 | −14 | −6 | 24 |
| Left ACC | 106 | 15.0247 | 0 | 22 | 36 |
| Left superior medial gyrus | 63 | 14.9261 | 2 | 48 | 32 |
| *TMS × Visual stimulus interaction* | | | | | |
| Right linual gyrus | 181 | 23.9332 | 22 | −90 | −6 |

Abbreviations: ACC, anterior cingulate cortex; EVA, early visual areas; IFG, inferior frontal gyrus; MNI, Montreal Neurological Institute; MT, medio-temporal; TMS, transcranial magnetic stimulation.

**Table 2. Local $\beta$ values (*post hoc* comparisons from TMS × Visual stimulus interaction).**

| | | *T*-values | Corrected *P*-values |
| --- | --- | --- | --- |
| noTMS, Static | earlyTMS, Static | −0.25 | 1.00 |
| | lateTMS, Static | −0.36 | 1.00 |
| | noTMS, Moving | 1.2 | 1.00 |
| | earlyTMS, Moving | −3.2 | 0.04* |
| | lateTMS, Moving | −1.8 | 0.7 |
| earlyTMS, Static | lateTMS, Static | −0.11 | 1.00 |
| | noTMS, Moving | 1.2 | 1.00 |
| | earlyTMS, Moving | −3.6 | 0.02* |
| | lateTMS, Moving | −1.5 | 1.00 |
| lateTMS, Static | noTMS, Moving | 1.3 | 1.00 |
| | earlyTMS, Moving | −2.8 | 0.09 |
| | lateTMS, Moving | −1.8 | 0.8 |
| noTMS, Moving | earlyTMS, Moving | −4.2 | 0.003 |
| | lateTMS, Moving | −2.8 | 0.09 |
| earlyTMS, Moving | lateTMS, Moving | 1.4 | 1.00 |

Abbreviation: TMS, transcranial magnetic stimulation. *significant ($P < 0.05$) t-tests

Table 3. MNI co-ordinates of the IC for TMS(EVA) and TMS(MT).

| Component | Cluster region | Cluster extent | Peak MNI co-ordinate | | |
|---|---|---|---|---|---|
| | | | x | y | z |
| **TMS(EVA)** | | | | | |
| **TMS-specific networks** | | | | | |
| IC1* | EVA (bilateral) | 1843 | 4 | −85 | 20 |
| IC3 | EVA (right hemisphere) | 1557 | 5 | −92 | 2 |
| IC7* | EVA (bilateral) | 1243 | −1 | −81 | 18 |
| IC8 | EVA (bilateral) | 1622 | −1 | −88 | 3 |
| **Other networks** | | | | | |
| IC4 | EVA (left hemisphere) | 1363 | −8 | −89 | 16 |
| IC5 | Extrastriate (bilateral) | 587 | 2 | −35 | 59 |
| IC6 | Medium frontal (bilateral) | 634 | −4 | 12 | 65 |
| **TMS(MT)** | | | | | |
| **TMS-specific networks** | | | | | |
| IC1* | Medial temporal lobe (right hemisphere) | 1343 | 48 | −70 | 12 |
| | Inferior temporal lobe (right hemisphere) | 187 | 60 | −68 | −6 |
| | IPS (right hemisphere) | 218 | 22 | −64 | 52 |
| | Medium frontal lobe (left hemisphere) | 138 | −4 | 58 | 58 |
| | Superior parietal lobe (left hemisphere) | 127 | −6 | −64 | 60 |
| IC2 | Medial temporal lobe (right hemisphere) | 348 | 43 | −73 | 16 |
| | Superior parietal lobe (right hemisphere) | 377 | 6 | 59 | 56 |
| IC8* | Medial temporal lobe (right hemisphere) | 687 | 39 | −63 | 15 |
| **Other networks** | | | | | |
| IC4 | EVA (right hemisphere) | 420 | −3 | −78 | 2 |
| IC5 | IPS (bilateral) | 722 | 31 | −68 | 55 |
| | Superior frontal gyrus (bilateral) | 683 | −13 | 69 | 12 |
| IC7 | Superior parietal lobe (bilateral) | 132 | −48 | −18 | 52 |
| IC6 | Precuneus (bilateral) | 456 | 4 | −64 | 58 |

Abbreviations: EVA, early visual areas; IC, independent component, IPS, intraparietal sulcus; MNI, Montreal Neurological Institute; MT, medio-temporal; TMS, transcranial magnetic stimulation. *IC significantly ($P < 0.05$) modulated by TMS conditions.

was no stimulation site effect, demonstrating that the local response to TMS was similar in both regions.

### Network changes in response to TMS perturbation

We further investigated whether and how long-range brain activity was modulated by TMS during MDD. We first applied an ICA to extract hidden spatiotemporal components contained within the TMS(EVA) and TMS(MT) task data, returning a measure of functional connectivity. We then applied temporal regression analysis on the components' time courses to compare network activity associated with the different experimental conditions.

The ICA applied to the TMS(EVA) data during motion discrimination revealed eight independent networks, of which seven were considered as functional brain networks. Five networks (IC1, IC3, IC4, IC7 and IC8) were labelled as TMS-related networks because they overlapped with the stimulation site (Fig. 4A and Table 3 for the MNI co-ordinates of the IC). IC1, IC7 and IC8 covered the bilateral EVA, whereas IC3 and IC4 encompassed the right EVA and left EVA, respectively. Only IC1 and

IC7 were significantly modulated by TMS, as revealed by one-way ANOVAs modelling the *t* scores obtained from the temporal regression analysis on the components' time courses for moving and static stimuli. IC1 was significantly downregulated by both TMS onsets when associated only with moving stimuli (one-way ANOVA: $F_{(2,15)} = 3.6$,

$P = 0.04$, *post hoc* TMS$_{early}$ *vs.* noTMS: $t_{(15)} = 1.9, P = 0.07$, TMS$_{late}$ *vs.* noTMS: $t_{(15)} = 2.2, P = 0.04$). IC7 exhibited a different pattern of modulation reflecting a significant overexpression of the bilateral EVA only during TMS$_{early}$, for moving ($F_{(2,15)} = 3.5$, $P = 0.045$, *post hoc* TMS$_{early}$ *vs.* noTMS: $t_{(15)} = 2.4, P = 0.03$) and static stimuli

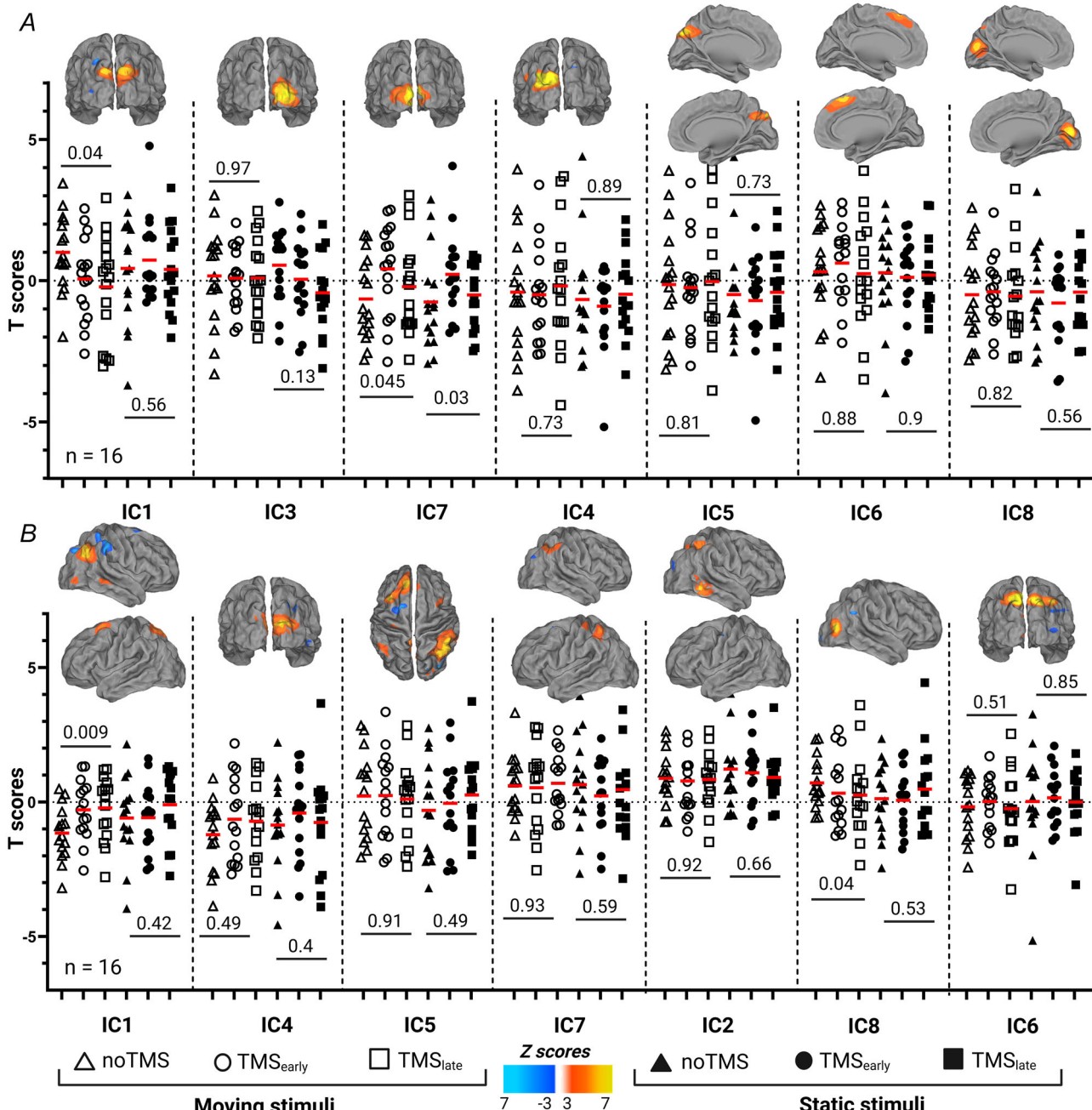

**Figure 4. TMS (transcranial magnetic stimulation)-induced network activity**
*A*, TMS$_{(EVA)}$-specific component networks that are modulated by TMS during motion discrimination and temporal regression analysis for the noTMS and TMS$_{(EVA)}$ conditions associated with moving or static stimuli (*n* = 16, one-way ANOVA); *B*, TMS$_{(MT)}$-specific component networks that are modulated by TMS during motion discrimination and temporal regression analysis for the noTMS and TMS$_{(MT)}$ conditions associated with moving or static stimuli (*n* = 16, one-way ANOVA); note that the components are sorted based on the percentage of explained variance.

($F_{(2,15)}$ = 4.4, $P$ = 0.03, *post hoc* TMS$_{early}$ *vs.* noTMS: $t_{(15)}$ = 2.3, $P$ = 0.035). All the other ANOVAs were not significant (see Fig. 4*A*).

When TMS was applied over the MT area, seven components were considered as functional brain networks. Among them three networks (IC1, IC2 and IC8) were labelled as TMS-related networks (see Table 3 for the MNI co-ordinates of the IC). Interestingly these networks were not restricted to only the right MT. IC2 and IC8 also included the IPS and superior parietal cortex and the medial frontal gyrus. IC1 activity was significantly enhanced by both TMS onsets when stimuli were moving ($F_{(2,15)}$ = 5.9, $P$ = 0.05, *post hoc* TMS$_{early}$ *vs.* noTMS: $t_{(15)}$ = 2.7, $P$ = 0.02, TMS$_{late}$ *vs.* noTMS: $t_{(15)}$ = 3.8, $P$ = 0.017). All the other ANOVAs were not significant (see Fig. 4*B*).

Finally we used graph-theoretical analysis to assess the topological and organizational properties of these networks while being agnostic to the TMS condition and visual stimulus. We first investigated small-world topological properties especially because networks with high small-world properties are believed to play a role in protecting the brain from local perturbation or damage. Both EVA and MT networks exhibited small-world properties ($\lambda \approx 1$ and $\sigma > 1$) among the selected sparsity values (see 'Methods' section). Interestingly there was a significant difference in the small-world network parameters (Sigma) between EVA networks and MT networks, with higher Sigma values for EVA networks at the sparsity values 0.3, 0.4 and 0.45 and marginally at 0.35 (Fig. 5*A*, left panel, and Table 4 for all paired *t* tests), also considering the mean across all sparsity values ($t_{(12)}$ = 2.4, $P$ = 0.034) (Fig. 5*B*, left panel). There was no difference in normalized clustering coefficient (Gamma) ($t_{(14)}$ = 0.42, $P$ = 0.68) (Fig. 5*A* and *B*, middle panels, and Table 4 for all paired *t* tests). Finally characteristic path length (Lambda) was significantly higher at sparsity thresholds of 0.2 and 0.25 and marginally at 0.15 for MT networks (Fig. 5*A*, right panel, and Table 4 for all paired *t* tests). The difference between EVA and MT networks was also significant when averaging across scarcities ($t_{(14)}$ = 2.4, $P$ = 0.03, Cohen's *d*: 0.75) (Fig. 5*B*, right panel).

## Predicting TMS effects on behaviour with BOLD activity

To explain the effect of TMS on MDD, we performed a series of multiple linear regressions using for each

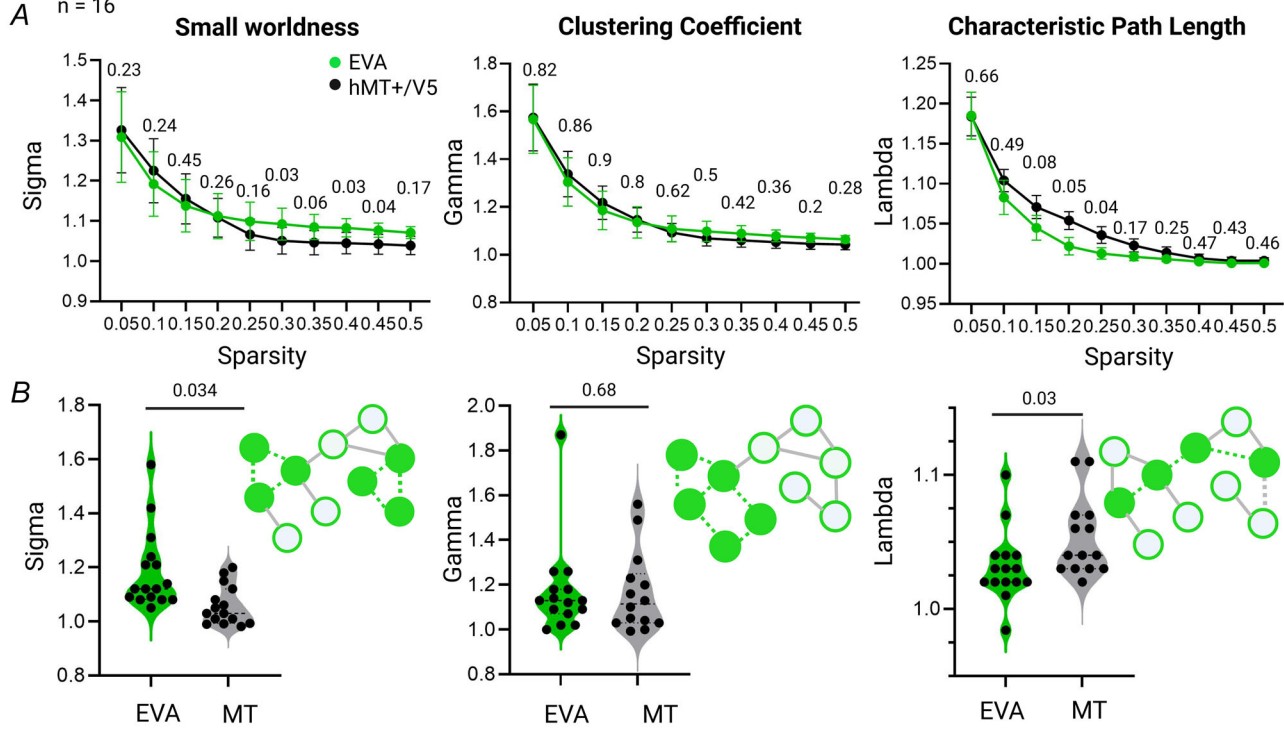

**Figure 5. TMS (transcranial magnetic stimulation)-induced network properties**
*A*, small-worldness properties (left panel), normalized clustering coefficients (middle panel) and normalized characteristic path length (right panel) for TMS$_{(EVA)}$ and TMS$_{(MT)}$ along the sparsity thresholds of 0.05–0.50; *B*, associated mean values of small-worldness properties (left panel), normalized clustering coefficients (middle panel) and normalized characteristic path length (right panel) for TMS$_{(EVA)}$ and TMS$_{(MT)}$ ($n$ = 16, paired *t* tests).

**Table 4. Paired *t* tests for the three topological markers between TMS$_{(EVA)}$ and TMS$_{(MT)}$ networks.**

| Sparsity thresholds | *t* | DF | *P*-value |
|---|---|---|---|
| **Difference in small-worldness (Sigma)** | | | |
| **0.05** | 0.084 | 14 | 0.23 |
| 0.1 | 0.121 | 14 | 0.24 |
| 0.15 | 0.31 | 14 | 0.45 |
| 0.2 | 0.52 | 14 | 0.26 |
| 0.25 | 0.84 | 14 | 0.16 |
| 0.3* | 2.01 | 14 | 0.03 |
| 0.35 | 1.89 | 14 | 0.06 |
| 0.4* | 2.01 | 14 | 0.03 |
| 0.45* | 2.02 | 14 | 0.04 |
| 0.5 | 1.45 | 14 | 0.17 |
| **Difference in clustering coefficient (Gamma)** | | | |
| 0.05 | 0.237 | 14 | 0.82 |
| 0.1 | 0.175 | 14 | 0.86 |
| 0.15 | 0.128 | 14 | 0.9 |
| 0.2 | 0.26 | 14 | 0.8 |
| 0.25 | 0.5 | 14 | 0.62 |
| 0.3 | 0.713 | 14 | 0.5 |
| 0.35 | 0.845 | 14 | 0.42 |
| 0.4 | 0.946 | 14 | 0.36 |
| 0.45 | 1.089 | 14 | 0.2 |
| 0.5 | 1.089 | 14 | 0.28 |
| **Difference in characteristic path length (Lambda)** | | | |
| 0.05 | 0.452 | 14 | 0.66 |
| 0.1 | −0.72 | 14 | 0.49 |
| 0.15(*) | −1.9 | 14 | 0.08 |
| 0.2* | −2.03 | 14 | 0.05 |
| 0.25* | −2.16 | 14 | 0.04 |
| 0.3 | −1.45 | 14 | 0.17 |
| 0.35 | −1.2 | 14 | 0.25 |
| 0.4 | −0.75 | 14 | 0.47 |
| 0.45 | −0.8 | 14 | 0.43 |
| 0.5 | −0.77 | 14 | 0.46 |

*Note*: D*f*, degree of freedom; *significant ($P < 0.05$) difference between EVA and MT networks.
Abbreviations: EVA, early visual areas; MT, medio-temporal; TMS, transcranial magnetic stimulation.

of the four conditions (TMS$_{(EVA)}$early, TMS$_{(EVA)}$late, TMS$_{(MT)}$early, TMS$_{(MT)}$late) the associated local $\beta$ scores, the *t* scores derived from the temporal regression of the significant TMS-ICs and the three graph theory metrics as fixed effects. The significant model explained the decrease in performance associated with TMS$_{(MT)}$early with Gamma values from the graph theory analysis as confirmed by ANOVA on the model's parameters ($F_{(1,6)} = 6.6$, $P = 0.04$), suggesting that TMS exerted a negative impact especially in participants exhibiting lower cluster coefficient, implying a more random or decentralized network structure. All other models were not significant (see Table 5 for all multiple linear regression results).

## Discussion

The aim of this study was to examine whether TMS perturbation elicits interindividual differences in BOLD responses, potentially reflecting the magnitude of the induced behavioural impairment. We used short bursts of 10 Hz TMS to disrupt either a low-level visual area (EVA) or a highly specialized, motion-sensitive, visual area (MT) during concurrent fMRI. We first confirmed the causal involvement of both regions in MDD, with more pronounced behavioural impairments after stimulation of MT. Our main findings were (1) a local, context-dependent increase in BOLD activity common to both targeted areas, (2) distinct patterns of network reconfiguration and (3) a relationship between TMS-induced performance decline after MT stimulation

**Table 5. Results of multiple regression analyses.**

| Models | ANOVA | SS | DF | MS | F(DFn, DFd) | *P*-value |
|---|---|---|---|---|---|---|
| TMS$_{(EVA)}$ early | Regression | 2834 | 6 | 472.4 | F(6,7) = 1.007 | P=0.4886 |
| | BetaV1 | 144.4 | 1 | 144.4 | F(1,7) = 0.3077 | P=0.5964 |
| | IC1 | 65.44 | 1 | 65.44 | F(1,7) = 0.1394 | P=0.7199 |
| | IC7 | 1999 | 1 | 1999 | F(1,7) = 4.260 | P=0.0779 |
| | meanLambda | 406.9 | 1 | 406.9 | F(1,7) = 0.8670 | P=0.3828 |
| | meanGamma | 58.02 | 1 | 58.02 | F(1,7) = 0.1236 | P=0.7355 |
| | MeanSigma | 2.066 | 1 | 2.066 | F(1,7) = 0.004402 | P=0.9490 |
| | Residual | 3285 | 7 | 469.3 | | |
| | Total | 6119 | 13 | | | |
| TMS$_{(MT)}$ early | Regression | 3287 | 6 | 547.9 | F(6,6) = 1.719 | P=0.2634 |
| | BetaMT | 2.931 | 1 | 2.931 | F(1,6) = 0.009196 | P=0.9267 |
| | IC1 | 350.6 | 1 | 350.6 | F(1,6) = 1.100 | P=0.3347 |
| | IC8 | 25.8 | 1 | 25.8 | F(1,6) = 0.08095 | P=0.7856 |
| | meanLambda | 0.08906 | 1 | 0.08906 | F(1,6) = 0.0002794 | P=0.9872 |
| | **meanGamma** | **2115** | **1** | **2115** | **F(1,6) = 6.636** | **P=0.0420\*** |
| | MeanSigma | 18.98 | 1 | 18.98 | F(1,6) = 0.05956 | P=0.8153 |
| | Residual | 1913 | 6 | 318.8 | | |
| | Total | 5200 | 12 | | | |
| TMS$_{(EVA)}$ late | Regression | 381.9 | 6 | 63.65 | F(6,4) = 5.470 | P=0.0610 |
| | BetaEVA | 4.257 | 1 | 4.257 | F(1,4) = 0.3659 | P=0.5779 |
| | IC7 | 48.24 | 1 | 48.24 | F(1,4) = 4.146 | P=0.1114 |
| | IC3 | 43.62 | 1 | 43.62 | F(1,4) = 3.749 | P=0.1249 |
| | meanLambda | 33.91 | 1 | 33.91 | F(1,4) = 2.915 | P=0.1630 |
| | meanGamma | 0.8721 | 1 | 0.8721 | F(1,4) = 0.07495 | P=0.7978 |
| | MeanSigma | 0.6260 | 1 | 0.6260 | F(1,4) = 0.05380 | P=0.8280 |
| | Residual | 46.54 | 4 | 11.64 | | |
| | Total | 428.4 | 10 | | | |
| TMS$_{(MT)}$ late | Regression | 334.7 | 6 | 55.78 | F(6,6) = 0.9866 | P=0.5063 |
| | BetaMT | 7.344 | 1 | 7.344 | F(1,6) = 0.1299 | P=0.7309 |
| | IC1 | 296.9 | 1 | 296.9 | F(1,6) = 5.250 | P=0.0618 |
| | IC8 | 132.4 | 1 | 132.4 | F(1,6) = 2.342 | P=0.1768 |
| | meanLambda | 88.09 | 1 | 88.09 | F(1,6) = 1.558 | P=0.2585 |
| | meanGamma | 51.87 | 1 | 51.87 | F(1,6) = 0.9174 | P=0.3751 |
| | MeanSigma | 16.00 | 1 | 16.00 | F(1,6) = 0.2830 | P=0.6139 |
| | Residual | 339.2 | 6 | 56.54 | | |
| | Total | 673.9 | 12 | | | |

*Note*: \*Significant (*P* < 0.05) model.
Abbreviations: DF, degree of freedom; EVA, early visual areas; IC, independent component; MS, mean square; MT, medio-temporal cortex; SS, sum of squares; TMS, transcranial magnetic stimulation.

and a shift toward a more random network structure. These results suggest that TMS perturbation triggers area-specific topological adaptations, which may reflect differing levels of network resilience to focal disruption.

### TMS-induced behavioural deterioration

In line with the concept of 'virtual lesion' described in the literature (Pascual-Leone et al., 1999; Beynel et al., 2019), TMS bursts applied to either EVA or MT resulted in a selective and transient deterioration of performance of a left/right MDD task (except for the late TMS$_{(EVA)}$ condition). This was interpreted to reflect successful causal and selective disruption of brain functioning. Interestingly participants were able to dissociate moving from static stimuli. This dissociation is likely supported by the existence of independent substrates mediating unspecific motion processing *versus* global motion direction integration in EVA and MT (Simoncelli & Heeger, 1998). There is evidence showing

that a subpopulation of neurons in MT focuses on the processing of local motion signals similar to that in EVA, and some other neuronal populations rather process global motion (Born & Tootell, 1992). Our TMS perturbation might have specifically affected the neurons involved in motion directions decoding. The absence of significant impairment induced by the late onsets of TMS$_{(EVA)}$ might be explained by differences in time delays of feedback loops in the motion discrimination system (Foss & Milton, 2000; Joukes et al., 2014). (Bridge et al., 2008; Ajina et al., 2015; Abed Rabbo et al., 2015).

### Local TMS-induced activity

Our results demonstrate that short bursts of TMS can elicit local BOLD activation beneath the coil in both EVA and MT, even at relatively low intensities – below the phosphene threshold. Although the presence of local BOLD activity elicitation in response to TMS is still unresolved in the field, we specifically found a local overactivation in presence of moving visual stimuli for both regions. The observed local positive BOLD underneath the coil paired with impaired motion performance can be interpreted in several, non-mutually exclusive ways. One possibility is that the positive BOLD reflects net excitation, meaning an overall increase in local synaptic or spiking activity that produces higher metabolic demand and thus increases BOLD signal (Howarth et al., 2020). However concurrent TMS–fMRI studies revealed a complex situation: although suprathreshold TMS to primary motor or visual cortex often produces local BOLD increases (likely driven by downstream motor or phosphene elicitation), TMS to many cortical targets does not reliably increase local BOLD at rest, and local BOLD effects appear strongly state dependent (Rafiei & Rahnev, 2021). In particular Rafiei and Rahnev (2022) argue that TMS can evoke alternating periods of increased and decreased firing that may cancel in terms of net metabolic signal, and that task engagement can unmask different haemodynamic outcomes (Rafiei & Rahnev, 2022).

An alternative account is that the positive BOLD reflects disinhibition (reduced local inhibitory drive) or a shift in excitation–inhibition balance: disinhibition can produce increased local firing and haemodynamic signals while simultaneously disrupting precise, information-bearing spiking patterns required for task performance. One can also speculate that this overactivity reflects a compensatory recruitment. TMS may perturb the targeted area and cause increased local BOLD because neighbouring or connected populations increase activity to compensate for perturbation, yet this compensatory activity is functionally inefficient and does not prevent behavioural impairment.

Finally this coupling between local BOLD increase and decreased performance suggests that the additional activity may reflect the accumulation of task-irrelevant neural noise (Ruff et al., 2009; Bancroft et al., 2014). Such a mechanism aligns with established models of TMS-induced disruption, where stimulation is thought to activate neuronal populations that are not optimally tuned to the task-relevant feature, in this case motion direction (Silvanto & Pascual-Leone, 2008). As a result the effective signal-to-noise ratio within the motion-sensitive network would be reduced, impairing accurate motion discrimination. Although we cannot firmly decide for one mechanism over the others, these results reinforce the idea that local BOLD changes after TMS can reflect mixed effects rather than a simple monotonic 'excitation' signal. Future work combining concurrent TMS with measures that index the excitation–inhibition balance (e.g. MR spectroscopy, laminar fMRI, functional Diffusion Weigthed Imaging (DWI) or invasive recordings) would help dissociate these possibilities.

Our results also highlight the strong state-dependent nature of TMS effects: stimulation during active motion direction processing leads to qualitatively different outcomes than during static stimuli presentation, suggesting that the excitability and engagement of the targeted network at the time of stimulation critically shape the neuronal consequences. These findings reinforce the idea that TMS does not simply 'excite' or 'inhibit' local tissue but interacts dynamically with ongoing neural activity (Romero et al., 2019; Siebner et al., 2022). This, together with other recent findings (Perera et al., 2024; Luo et al., 2025), might open the door to more tailored or context-sensitive brain stimulation protocols. Beyond the active/rest comparison we showed that TMS interacts with the ongoing neural context, such that stimulation timing relative to task processing stages determines both behavioural consequences and downstream network responses. Early stimulation of EVA and MT selectively impaired direction discrimination and induced distinct patterns of network reorganization, whereas later stimulation had no effect on EVA. This temporal specificity is in line with recent chronometric TMS–fMRI studies (Grosshagauer et al., 2024). Together these findings reinforce the principle that TMS should be understood as a state-dependent perturbation whose causal impact is dynamically shaped by neural processing and behavioural context.

### TMS-induced whole-brain activity

An important question is whether this 'local noise injection' propagates in the brain to destabilize larger functional networks. TMS-induced changes in BOLD activity were observed not only locally in the stimulated

regions but also in remote areas through functional coupling. For both EVA and MT stimulation, the spread of activity to distal regions – such as the IPS, frontal eye fields (FEF) and medial prefrontal cortex – replicates previous findings from TMS–fMRI studies (Ruff et al., 2006; Caparelli et al., 2010; Leitão et al., 2015). Particularly many of these remote regions are task relevant and are known to be engaged during motion perception and direction discrimination (Offen et al., 2010; Sani et al., 2021), suggesting that the ongoing task context may also shape the propagation of TMS-evoked activity through the actively engaged network.

When TMS was applied during motion processing, additional BOLD activation was observed in regions such as the MT cortex and medial frontal areas. Importantly this increased activation coincided with a measurable decline in behavioural performance on the MDD task, as well as with participants' subjective reports of increased task difficulty. These findings suggest that TMS may disrupt efficient network integration during perceptual processing, potentially by altering the balance of activity within task-relevant networks. The differential effect induced by the early and late TMS onsets was visible only when EVA was stimulated, and the effect manifested as an increase in activity in the right FEF and in the medial prefrontal cortex with the late onsets. These regions are involved in perceptual decision-making, and this time could reflect a compensatory mechanism that maintains stable performance (Imani et al., 2021).

## TMS-induced network activity

To provide both a qualitative and quantitative framework for comparing network activity across different TMS-induced perturbations, we combined group ICA with graph-theoretical metrics. ICA identified multiple functional networks, some directly linked to TMS and others more domain general. Across both targeted regions (EVA and MT), the decomposition revealed components with divergent responses to TMS. In the case of EVA, for example, IC7 – spatially overlapping with the site of stimulation – exhibited increased activation specifically after early TMS onset. In contrast IC1, located more rostrally, exhibited a pattern of downregulation under the same stimulation condition. The opposing responses may reflect a balance between excitatory and inhibitory processes, possibly mediated by intra-areal or feedback projections (Shao & Burkhalter, 1996; Schwabe et al., 2006). Furthermore the specificity of IC7's response to early TMS onset raises questions about the temporal dynamics of network susceptibility, with different networks or nodes having critical windows of 'vulnerability' or influence (Huang & Yu, 2017; Kottaram et al., 2018). This dissociation suggests that

ICA successfully captured the complex, spatially and temporally differentiated dynamics of TMS-induced responses. It highlights ICA's sensitivity not only to spatial localization but also to the functional state of the network – whether it is task relevant, engaged or modulated by TMS in a context-dependent manner. It further emphasizes that TMS effects are not uniform within a target region but depend on the local functional architecture and current cognitive or perceptual state.

Regarding network activity induced by $TMS_{(MT)}$, we found that TMS engaged more spatially distributed networks, involving additional associative regions such as the FEF, the posterior parietal cortex (PPC) and the medial prefrontal cortex (mPFC). In contrast all but one of the ICs associated with $TMS_{(EVA)}$ were largely confined to the stimulated visual area, indicating more localized network engagement. To quantitatively assess these differences we applied graph-theoretical analyses and observed distinct whole-brain network configurations depending on the stimulation site. Specifically networks following TMS over EVA exhibited higher small-worldness – a topological property characterized by strong local clustering and short path lengths – compared to those following TMS over MT. This network architecture is thought to support efficient parallel information processing by maximizing information transfer at minimal wiring cost (Achard & Bullmore, 2007). The clustered, locally efficient organization of the $TMS_{(EVA)}$ network may promote functional resilience and flexible redistribution of activity. This could help explain the relative preservation of behavioural performance during late TMS onset, suggesting that the network maintains coherence despite transient perturbations (Wu et al., 2020). In contrast the more dispersed $TMS_{(MT)}$ network configuration likely reflects broader propagation of task-specific interference across higher-order visual regions, potentially leading to a less-efficient, more vulnerable network state. Congruently the variable that best accounted for the TMS-induced performance impairments after $TMS_{(MT)}$ was the clustering coefficient. This measure reflects the tendency of nodes within a network to form tightly interconnected clusters. High clustering values indicate a well-organized, locally efficient network structure, whereas lower values reflect a more random or fragmented configuration, consistent with patterns observed in our ICA analyses. Our findings showed that participants who exhibited larger performance declines had lower clustering coefficients, suggesting that a more dispersed network organization is detrimental to MDD. It would be valuable in future work to investigate whether these network disruptions extend to other visual sub-functions – or even domain-general cognitive functions – to determine the broader functional impact of focal perturbations on brain-wide network dynamics. Interestingly in many domains of network science, systems

with higher local clustering and small-world properties are more resilient to perturbation because they enable redundant, short-range communication paths (e.g. Eom, 2018). In the brain similar principles have been observed in studies of lesion impact, epilepsy and ageing, where disruption of highly distributed hubs often has stronger behavioural consequences than perturbation of more locally clustered regions (Reber et al., 2021; Hwang et al., 2021).

This result fits with our initial hypothesis, which was that EVA, due to its role as an early-stage integrative cortical area, would maintain more stable synaptic and network dynamics in response to TMS, resulting in greater resistance to externally induced disruptions in both perceptual performance and functional connectivity compared to MT. It is in line with a modelling study, showing that brain hubs, where activity is integrated and further distributed (like EVA in our case), operate in a slower regime and appear to be functionally less affected by focal perturbations (Gollo et al., 2017). In contrast perturbations of peripheral or more associative areas (e.g. MT) might have greater impact on acute network activity. Lesion studies also provided evidence that damage to the homologue of MT in behaving cats impairs more relearning of motion discrimination and learning transfer than lesions to EVA (Das et al., 2012). In sum the network analyses revealed differences between the two disrupted areas, not observable with classical activation analyses. It highlights the relevance of graph-theoretical metrics in capturing functionally meaningful changes in network topology. In particular the clustering coefficient appears to be a sensitive marker of TMS-induced network reconfiguration and its behavioural consequences. More globally these results support the notion that topology-based markers can predict which nodes are 'fragile' *versus* 'robust' under external disruption, strengthening the contribution of this work to the emerging framework of causal disconnectomics.

## Limitations of the current work and open questions

The aim of this study was to develop a causal connectomic framework for understanding MDD by selectively perturbing two key network nodes using TMS. It is important to observe that direct translation of our results to brain injuries is not straightforward. Unlike brain lesions TMS-induced perturbations are both temporally transient and spatially focal. More critically the physiological consequences of a lesion might differ fundamentally from those of TMS. Brain lesions not only disrupt local neuronal activity but also provoke widespread, often non-specific responses – particularly in the hyperacute phase – such as excitotoxicity, inflammation and diaschisis (Lai et al., 2014; Farooqui

et al., 2022). These processes can profoundly alter inter-regional communication through mechanisms unrelated to normal synaptic signalling.

In contrast the type of TMS used in this study induces neuronal firing and interareal communication, though likely mediated by different – and possibly more superficial or excitatory – neuronal populations than those affected by a brain lesion. This distinction is essential when interpreting the resulting BOLD signal changes. Although increases in BOLD activity are often taken to reflect compensatory engagement, they could alternatively indicate maladaptive or inefficient network recruitment. Future work using population tuning models or decoding approaches could help determine the functional relevance of these activations and determine whether they truly reflect adaptive reorganization or instead a signature of disrupted processing.

Another intriguing aspect concerns the resting-state TMS–fMRI data (presented in the supplementary results). These results helped confirm the spatial specificity of the TMS effects, validating that stimulation targeted the intended network nodes even at rest. However we observed that the direction of BOLD signal changes at rest contrary to those seen during task performance. This apparent discrepancy has been reported in prior studies and may reflect state-dependent effects of TMS; that is the neural response to TMS can differ significantly depending on whether the brain is at rest or engaged in a task. Another contributing factor could be the experimental design itself: we employed a block design during the resting-state acquisition and an event-related design during the task-based sessions. These methodological differences could influence the temporal integration of BOLD responses and the observed directionality of signal change (Chee et al., 2003; Tie et al., 2009). Together these findings underscore the complexity of interpreting TMS-induced effects in fMRI data and highlight the need to consider both brain state and experimental context when drawing causal inferences. Another important consideration for future studies is the inclusion of additional control baselines, such as sham TMS or stimulation of a control site during fMRI. These approaches could provide further leverage to dissociate specific network effects of TMS from non-specific auditory or somatosensory contributions.

## Conclusion

Despite growing interest and significant advances in computational modelling, the causal mechanisms underlying neural network dynamics remain incompletely understood. In this study we employed concurrent TMS–fMRI to probe the neural mechanisms of large-scale network adaptation using a targeted perturbation approach.

Our findings offer neuroimaging evidence for the context-dependent nature of TMS effects, suggesting that such perturbations may preferentially engage specific neuronal populations depending on task demands. Moreover we show that inducing focal perturbations at different levels of the visual hierarchy leads to distinct patterns of dysregulation across visual networks.

Complementing classical lesion studies (Das et al., 2012) our results highlight the utility of TMS–fMRI coupling as a powerful tool for investigating 'causal disconnectomics', the causal relationships between localized disruptions and large-scale neural and behavioural consequences (Egger et al., 2021). This approach holds promise for precisely mapping how local perturbations propagate through brain networks, especially when assessed during active cognitive processing. Ultimately such techniques may advance our understanding of brain resilience, compensation and vulnerability in the face of focal damage.

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

## Additional information

### Data availability statement

The analytic codes for this study are available in Zenodo at DOI 10.5281/zenodo.17511167. The data supporting the findings of the present study are available from the corresponding author upon reasonable request. Access will be granted in compliance with ethical and privacy regulations and only for purposes that align with the study aims and participants' consent.

### Competing interests

The authors declare that they have no financial or personal conflicts of interest related to the subject matter of this study.

### Author contributions

E.R. designed the research work, conducted research, analysed data, wrote and revised the manuscript and secured funding. F.C.H. designed the research work, discussed the data, revised the manuscript and secured funding. R.F.S.-G. and K.R.H. assisted with the behavioural experimental scripts and revised the manuscript. O.R. provided support for the experimental script used in concurrent TMS–fMRI data acquisition and assisted with data acquisition. L.M. and R.M. contributed to the operation involving the TMS–fMRI acquisition. All authors have approved the final version of the manuscript submitted for publication and agree to be accountable for all aspects of the work. All persons designated as authors qualify for authorship, and all those who qualify for authorship are listed.

### Funding

This study was supported by the Bertarelli Foundation (Catalyst BC77O7 to F.C.H. and E.R.), the Swiss National Science Foundation (PRIMA PR00P3_179867 to E.R.) and the Defitech Foundation (to F.C.H.).

### Acknowledgements

The authors thank Holly Bridge for her insightful comments on the manuscript. The authors thank the MRI and neuro-modulation facilities of the Human Neuroscience Platform of the Fondation Campus Biotech Geneva for technical advice and technical help during data acquisition.

Open access publishing facilitated by Ecole polytechnique federale de Lausanne, as part of the Wiley - Ecole polytechnique federale de Lausanne agreement via the Consortium Of Swiss Academic Libraries.

**Keywords**

functional disconnectomics, motion direction discrimination, network analyses, online TMS–functional magnetic resonance imaging (fMRI), transcranial magnetic stimulation (TMS) perturbation

**Supporting information**

Additional supporting information can be found online in the Supporting Information section at the end of the HTML view of the article. Supporting information files available:

**Peer Review History**

