## [Peer Review History · The Journal of Physiology]

Causal disconnectomics of motion perception networks: insights from TMS-induced BOLD responses

Estelle Raffin, Roberto Felipe Salamanca-Giron, Krystel R Huxlin, Olivier Reynaud, Loan Mattera, Roberto Martuzzi, and Friedhelm Hummel

DOI: 10.1113/JP289699

Corresponding author(s): Estelle Raffin (estelle.raffin@univ-grenoble-alpes.fr)

Review Timeline:

Submission Date:	11-Jul-2025
Editorial Decision:	15-Sep-2025
Revision Received:	29-Sep-2025
Editorial Decision:	21-Oct-2025
Revision Received:	03-Nov-2025
Accepted:	04-Nov-2025

Senior Editor: Richard Carson

Reviewing Editor: Bettina Schwab

Transaction Report:

Dear Dr Raffin,

Re: JP-RP-2025-289699 "**Causal disconnectomics of motion perception: insights from TMS-induced BOLD responses**" by Estelle Raffin, Roberto Felipe Salamanca-Giron, Krystel Huxlin, Olivier Reynaud, Loan Mattera, Roberto Martuzzi, and Friedhelm Hummel

Thank you for submitting your manuscript to The Journal of Physiology. It has been assessed by a Reviewing Editor and by 2 expert referees and we are pleased to tell you that it is potentially acceptable for publication following satisfactory major revision.

LANGUAGE EDITING AND SUPPORT FOR PUBLICATION: If you would like help with English language editing, or other article preparation support, Wiley Editing Services offers expert help, including English Language Editing, as well as translation, manuscript formatting, and figure formatting at www.wileyauthors.com/eoo/preparation. You can also find resources for Preparing Your Article for general guidance about writing and preparing your manuscript at www.wileyauthors.com/eoo/prepresources.

REVISION CHECKLIST:

We look forward to receiving your revised submission.

Yours sincerely,

Richard Carson
Senior Editor
The Journal of Physiology

REQUIRED ITEMS

- 1) - Include a Key Points list in the article itself, before the Abstract.
 - 2) - Author photo and profile. First or joint first authors are asked to provide a short biography (no more than 100 words for one author or 150 words in total for joint first authors) and a portrait photograph. These should be uploaded and clearly labelled together in a Word document with the revised version of the manuscript. See Information for Authors for further details.
 - 3) - Your manuscript must include a complete Additional Information section, including competing interests; funding; author contributions and acknowledgements.
 - 4) - Please upload separate high-quality figure files via the submission form.
 - 5) - Your paper contains Supporting Information of a type that we no longer publish, including supplementary tables and figures. Any information essential to an understanding of the paper must be included as part of the main manuscript and figures. The only Supporting Information that we publish are video and audio, 3D structures, program codes and large data files. Your revised paper will be returned to you if it does not adhere to our Supporting Information Guidelines.
 - 6) - Papers must comply with the Statistics Policy: https://jp.msubmit.net/cgi-bin/main.plex?form_type=display_requirements#statistics.
- In summary:
- If $n \leq 30$, all data points must be plotted in the figure in a way that reveals their range and distribution. A bar graph with data points overlaid, a box and whisker plot or a violin plot (preferably with data points included) are acceptable formats.
 - If $n > 30$, then the entire raw dataset must be made available either as supporting information, or hosted on a not-for-profit repository, e.g. FigShare, with access details provided in the manuscript.
 - 'n' clearly defined (e.g. x cells from y slices in z animals) in the Methods. Authors should be mindful of pseudoreplication.
 - All relevant 'n' values must be clearly stated in the main text, figures and tables.
 - The most appropriate summary statistic (e.g. mean or median and standard deviation) must be used. Standard Error of the Mean (SEM) alone is not permitted.
 - Exact p values must be stated. Authors must not use 'greater than' or 'less than'. Exact p values must be stated to three significant figures even when 'no statistical significance' is claimed.

7) - A Data Availability Statement is required for all papers reporting original data. This must be in the Additional Information section of the manuscript itself. It must have the paragraph heading 'Data Availability Statement'. All data supporting the results in the paper must be either: in the paper itself; uploaded as Supporting Information for Online Publication; or archived in an appropriate public repository. The statement needs to describe the availability or the absence of shared data. Authors must include in their statement: a link to the repository they have used, or a statement that it is available as Supporting Information; reference the data in the appropriate section(s) of their manuscript; and cite the data they have shared in the References section. Whenever possible, the scripts and other artefacts used to generate the analyses presented in the paper should also be publicly archived. If sharing data compromises ethical standards or legal requirements then authors are not expected to share it, but must note this in their statement. For more information, see our Statistics Policy.

8) - Please include an Abstract Figure file, as well as the Figure Legend text within the main article file. The Abstract Figure is a piece of artwork designed to give readers an immediate understanding of the research and should summarise the main conclusions. If possible, the image should be easily 'readable' from left to right or top to bottom. It should show the physiological relevance of the manuscript so readers can assess the importance and content of its findings. Abstract Figures should not merely recapitulate other figures in the manuscript. Please try to keep the diagram as simple as possible and without superfluous information that may distract from the main conclusion(s). Abstract Figures must be provided by authors no later than the revised manuscript stage and should be uploaded as a separate file during online submission labelled as File Type 'Abstract Figure'. Please also ensure that you include the figure legend in the main article file. All Abstract Figures should be created using BioRender. Authors should use The Journal's premium BioRender account to export high-resolution images. Details on how to use and access the premium account are included as part of this email.

9) - Please ensure that all figures and tables have a title and legend, and that they have been cited within the main article text.

EDITOR COMMENTS

Reviewing Editor:

Thank you for submitting your work to The Journal of Physiology. It was been reviewed by two experts in the field, who both appreciate the choice of the topic, and the robust and original work. Nevertheless, they also raised several concerns that all need to be addressed. In particular, the brain map figures require quantitative measures (e.g. t- or p-values). Please ensure that the data are presented transparently throughout the manuscript, including publicly available data and code. Furthermore, the translation to stroke, as mentioned in the abstract, is not evident.

- Fig. 4: provide data points or provide data as supporting information.
- brain map figures require quantitative measures (e.g. t- or p-values).
- data availability statement missing: please make data & code available publicly.

Senior Editor:

Please see the comments provided by the referees and the Reviewing Editor concerning the requirement of the Journal that data be presented as completely and as transparently as possible.

Please note that the Journal does not publish supplementary materials. All relevant information should be contained in the main body of the text.

REFeree COMMENTS

Referee #2:

In this manuscript, the authors delivered short bursts of 10 Hz TMS over either early visual areas (EVA) or the medio-temporal area (MT) in healthy participants, while acquiring concurrent functional MRI. The authors found that TMS delivered during the early stages of motion processing selectively impaired direction discrimination at both sites, while global motion perception remained unaffected.

The topic presented in this manuscript is intriguing. I appreciate the data collection and fMRI analysis. However, I have some conceptual questions that have to be addressed before the article is accepted for publication.

1. I found the title not very suitable for this manuscript. In the title and elsewhere, they talked about disconnections and claimed that their findings can be used to understand lesion-induced brain changes in neurological disorders such as stroke. However, this argument is not properly supported. In my opinion, this manuscript does not cover aspects of disconnection that can be translated to stroke patients. It is well known that spatial neglect is one of the most common behavioral consequences after brain injury, affecting predominantly the right hemisphere. Also, spatial neglect affects tasks of spatial attention and orientation, and patients typically deviate towards the ipsilesional side, neglecting contralesional information or stimuli. I think the authors have to elaborate more on the disconnection aspects.

2. The abstract needs to be rewritten. The main message of the manuscript is not clear.

3. The TMS protocol used in this study needs more explanation. The authors used short bursts of 10 Hz TMS, but they do not explain the rationale for using this approach. I would have expected the use of an inhibitory protocol instead of a 10 Hz protocol if they wanted to induce a disconnection-like effect. Please elaborate.

4. Lines 364 - 368. Please replace "informax" with "infomax". The authors use ICA to identify the networks of interest. They also used PCA to reduce the dimensionality of their data. Can the authors please specify the average number of dimensions to which the data were reduced using PCs, and the average number of ICs they sought? This information is relevant.

5. Section 3.3, lines 505 - 529. The authors used the IC number to indicate the networks of interest. See, for example, lines 507-509, "Five networks (IC1, IC3, IC4, IC7, and IC8) were labelled as TMS-related networks because they overlapped with the stimulation site (Figure 4A and Supplementary Table S5 for the MNI coordinates of the IC)." Then the reader has to wait until the supplementary material to see which networks the ICs represent. I suggest moving those tables to the main text or including more information about the networks in that section. Fig. 4 also should contain that information.

6. Overall, the literature review should be deepened, and more complete original papers should be added to the main manuscript where needed. See, for instance, <https://doi.org/10.1038/s41596-025-01182-4>.

7. Please add to Figure caption 1.A. that the coil was placed in the right EVA and right MT. Also, check the spelling in line 232. I think you meant "coil" instead of "oil".

I hope the authors find my comments helpful to improve the impact of the manuscript.

Referee #3:

This is a well-designed and timely study that has the potential to advance our understanding of how transient perturbation of different visual areas by TMS influences local BOLD activity, network organization, and motion perception behavior. The combination of TMS bursts well timed with ongoing task processing with concurrent fMRI as readout allows insights into what the authors call "causal disconnectomics." Its really hard to interpret the findings since even though we have a causal tool at hand, the BOLD changes could be interpreted in a "virtual lesion" framework, as compensatory effects, hyperactivation etc... The authors make a good job in keeping interpretation balances. The manuscript is clearly written and represents a strong contribution to the field. I offer the following comments for improvement:

Major:

Figures - color bars are missing: A major methodological concern is that the statistical brain map figures lack color bars. Without color bars, readers cannot interpret the magnitude or statistical thresholds of the activations, and the images remain only illustrative. Properly labeled color bars (e.g., mapping colors to t-values or p-values) should be included in all statistical figures to make the data interpretable and reproducible.

Minor:

Stimulation timing and state-dependency: The chronometric design is excellent, and the early vs late TMS effects are a strength. Results highlight that stimulation timing relative to neural processing stages is crucial. This ties directly into the principle of state-dependency-TMS interacts with ongoing neural activity and task engagement, not simply exerting uniform "excitatory" or "inhibitory" effects. It would be valuable to elaborate this point more explicitly in the Discussion. For context, recent chronometric TMS-fMRI studies (e.g., Grosshagauer et al., 2024) have shown that downstream BOLD responses differ depending on whether stimulation is delivered during different phases in task and at rest. This comparison with other literature could make state-dependent TMS effects more integrated.

Control conditions: The inclusion of noTMS and control-site (M1) conditions is a strong aspect of the study. It could still be helpful for the authors to discuss whether sham TMS or control-site stimulation during fMRI might serve as additional baselines in future work, especially to control for potential nonspecific auditory or somatosensory effects. In the current study this would be beyond the scope though.

Interpretation of local BOLD responses: The observed local BOLD "over-activation" beneath the coil during motion trials,

coupled with behavioral impairment. This pattern is compatible with models of TMS adding "neural noise" that reduces signal-to-noise ratio in task-relevant networks. I suggest the authors expand the Discussion on what positive BOLD increases mean physiologically: whether they reflect excitation, disinhibition, or compensatory activity. Referencing the debate in the literature (e.g., Rafiei & Rahnev, 2022) will strengthen this interpretation.

Network-level analysis and causal disconnectomics: The ICA and graph-theoretical results are compelling, showing distinct network reorganizations after EVA vs MT stimulation. The finding that clustering coefficient predicts behavioral impairment following MT stimulation is especially novel and should be emphasized. This link between topology and behavioral vulnerability is an important contribution to causal disconnectomics. It would also be valuable to discuss why EVA's network configuration might be more resilient (higher small-worldness, local clustering) compared to MT's more distributed pattern.

Broader implications: The study nicely connects experimental perturbations to concepts of network resilience and compensation. Another suggestion would be to highlight implications beyond vision—for example, whether similar principles may apply in clinical contexts such as stroke or depression, where causal mapping of network vulnerabilities could guide interventions.

END OF COMMENTS

Reviewing Editor:

Thank you for submitting your work to The Journal of Physiology. It was been reviewed by two experts in the field, who both appreciate the choice of the topic, and the robust and original work. Nevertheless, they also raised several concerns that all need to be addressed. In particular, the brain map figures require quantitative measures (e.g. t- or p-values). Please ensure that the data are presented transparently throughout the manuscript, including publicly available data and code. Furthermore, the translation to stroke, as mentioned in the abstract, is not evident.

Thank you for the positive anticipation of our manuscript. We have addressed all of these points. Please find our answers below.

- Fig. 4: provide data points or provide data as supporting information.

Individual data points are now provided in the new Figure 4.

Please note that we slightly changed Figure 2 to display in the first left column, the TMS x Site interaction for motion perception and motion discrimination with the associated p values. The associated posthoc comparisons are displayed with individual values and the p values associated with the t-scores are provided when the interaction was significant (for motion discrimination only):

Similarly, Figure 3G was slightly changed to display the TMS by Visual stimuli interaction and the associated p values instead of the triple interaction (TMS x Visual stimuli x Site) which was not significant, and from which no further posthoc results could be displayed on the figure.

- brain map figures require quantitative measures (e.g. t- or p-values).

We apologize for the omission, Z scores as well as the associated p values have been added to the figures and the captions.

- data availability statement missing: please make data & code available publicly.

The statement has been added to the manuscript

Senior Editor:

Please see the comments provided by the referees and the Reviewing Editor concerning the requirement of the Journal that data be presented as completely and as transparently as possible.

Please note that the Journal does not publish supplementary materials. All relevant information should be contained in the main body of the text.

We have revised the manuscript accordingly. Some methodological details and the necessary tables have been incorporated to the main text while some additional behavioral analyses, which were not essential to the main results have been discarded (e.g., old Supplementary Figure S1: Relationship between awareness and accuracy). Finally, some parts of the supplementary results are now openly provided in a public repository Zenodo at <https://zenodo.org/records/17175650>.

REFEREE COMMENTS

Referee #2:

In this manuscript, the authors delivered short bursts of 10 Hz TMS over either early visual areas (EVA) or the medio-temporal area (MT) in healthy participants, while acquiring concurrent functional MRI. The authors found that TMS delivered during the early stages of motion processing selectively impaired direction discrimination at both sites, while global motion perception remained unaffected.

The topic presented in this manuscript is intriguing. I appreciate the data collection and fMRI analysis. However, I have some conceptual questions that have to be addressed before the article is accepted for publication.

We thank reviewer #2 for his/her positive evaluation of our work. Please find below a point-to-point answer to all comments.

1. I found the title not very suitable for this manuscript. In the title and elsewhere, they talked about disconnections and claimed that their findings can be used to understand lesion-induced brain changes in neurological disorders such as stroke. However, this argument is not properly supported. In my opinion, this manuscript does not cover aspects of disconnection that can be translated to stroke patients. It is well known that spatial neglect is one of the most common behavioral consequences after brain injury, affecting predominantly the right hemisphere. Also, spatial neglect affects tasks of spatial attention and orientation, and patients typically deviate towards the ipsilesional side, neglecting contralesional information or stimuli. I think the authors have to elaborate more on the disconnection aspects.

We thank the reviewer for raising this important point. We agree that the direct translation of our results to stroke-induced lesions is not straightforward, and we did not intend to suggest that our study models common post-stroke syndromes such as spatial neglect. As the reviewer correctly

notes, spatial neglect typically follows damage to the right temporo-parietal junction and posterior parietal cortex and manifests as a disorder of spatial attention and orientation. By contrast, in the present study we specifically targeted motion discrimination capacities by perturbing relatively low-level visual areas (EVA and MT) during a motion discrimination task. While the motion discrimination network has been extensively characterized in humans and non-human primates, and may hold some clinical interest in the context of visual rehabilitation after occipital stroke (Das et al., 2014), we recognize that lesions specifically and selectively affecting this network are uncommon in clinical practice.

Our rationale for using this model was not to mimic stroke-induced syndromes directly, but rather to test the broader concept of *causal disconnectomics* under highly controlled experimental conditions. The recently introduced term *causal disconnectomics* (Glick et al., 2024) refers to the exploration of how focal perturbations can lead to large-scale reconfigurations of functional brain networks and their behavioral consequences. In this sense, our study provides a proof-of-principle demonstration: we show that perturbations of different nodes within a well-defined brain network yield dissociable patterns of network reorganization and behavioral effects. We believe this approach is valuable because it allows one to investigate fundamental principles of brain resilience, compensation, and vulnerability to focal disruptions, which in turn can inform hypotheses about lesion-induced reorganization in clinical populations.

We have revised the text to indicate that our findings do not directly model stroke symptoms such as neglect, but they illustrate the potential of TMS-fMRI coupling as a causal mapping tool that can eventually be applied to networks more directly implicated in neurological disorders, line 65:

“A central challenge in systems neuroscience is to understand how flexible and distributed brain networks give rise to complex behavior. While advances in neuroimaging have provided detailed descriptions of functional and structural brain connectivity (Blanco et al., 2024; Lim et al., 2019; Litwińczuk et al., 2022; Ma et al., 2022), these correlational approaches offer limited insights into the causal mechanisms that underlie network dynamics and behavior, resulting in an incomplete account of how focal perturbations causally reshape large-scale network dynamics.”

Line 114:

“This approach may ultimately allow us to characterize individual-specific network vulnerabilities that shape multidomain interactions, offering insights into how sensory, cognitive, and motor processes emerge from shared and distributed neural dynamics”

Note that we kept the paragraph in the discussion where the limitation of such approach for direct clinical translation was addressed line 744:

“It is important to bear in mind that the direct translation of our results to brain injuries is not straightforward. Unlike brain lesions, TMS-induced perturbations are both temporally transient and spatially focal. More critically, the physiological consequences of a lesion might differ fundamentally from those of TMS. Brain lesions not only disrupt local neuronal activity but also provoke widespread, often nonspecific responses—particularly in the hyperacute phase—such as excitotoxicity, inflammation, and diaschisis

(Farooqui et al., 2022; Lai et al., 2014). These processes can profoundly alter interregional communication through mechanisms unrelated to normal synaptic signaling.”

In the discussion, line 782:

“Despite growing interest and significant advances in computational modelling, the causal mechanisms underlying neural network dynamics remain incompletely understood. In this study, we employed concurrent TMS-fMRI to probe the neural mechanisms of large-scale network adaptation, using a targeted perturbation approach.

After a better explanation of the broader concept of causal disconnectomics throughout the paper, we would prefer keeping the title as it is: “Causal disconnectomics of motion perception networks: insights from TMS-induced BOLD responses”, but we are happy to further discuss with the reviewer if he/she thinks it is still not appropriate.

2. The abstract needs to be rewritten. The main message of the manuscript is not clear.

We have extensively rephrased the abstract to improve the clarity and we have removed any mention of direct translation to stroke-induced deficits. Here is the revised version:

“Understanding how focal perturbations trigger large-scale network reorganization is essential for uncovering the neural mechanisms that support perception and behavior. Here, we used a transcranial magnetic stimulation (TMS) perturbational approach by applying brief 10 Hz TMS to early visual areas (EVA) or the medio-temporal area (MT) in healthy participants while recording concurrent fMRI. TMS delivered during early stages of motion processing specifically impaired direction discrimination at both sites, while the disruption of the later processing phase only impaired performances for the MT condition. Despite a similar local increase in BOLD activity induced by EVA and MT stimulation, the broader network responses diverged markedly. Perturbation of EVA elicited a more robust and efficient pattern of functional reorganization, manifesting as more constrained BOLD changes, consistent with greater resilience to focal disruption. In contrast, behavioral impairments induced by MT stimulation were accompanied by a disorganized and less efficient network configuration, characterized by smaller small-world properties and longer path lengths. The drop in performances induced by MT stimulation scaled with lower clustering coefficients, implying a more random or decentralized network structure. These findings demonstrate that TMS-fMRI coupling provides a powerful framework for causally mapping the relationships between local neural perturbations, large-scale network dynamics, and behavioral performance.”

3. The TMS protocol used in this study needs more explanation. The authors used short bursts of 10 Hz TMS, but they do not explain the rationale for using this approach. I would have expected the use of an inhibitory protocol instead of a 10 Hz protocol if they wanted to induce a disconnection-like effect. Please elaborate.

The reviewer raises an interesting point. We deliberately chose an *online* perturbational approach with short bursts of 10 Hz TMS rather than an *offline* protocol such as iTBS or low-frequency rTMS, which are typically designed to induce longer-lasting changes in cortical excitability. This was motivated by several factors: 1) Online bursts allow trial-by-trial perturbation, enabling randomization across multiple stimulation conditions within the same session. This results in a cleaner design and reduces confounds related to session order or inter-individual variability in offline stimulation effects; 2) online stimulation provides direct access to the chronometry of neural processing, allowing us to probe the causal contribution of a targeted region at specific stages of motion discrimination. Such temporal precision cannot be achieved with offline protocols, which modulate excitability for minutes but without control over task timing; 3) our primary aim was to investigate how focal perturbations dynamically reshape large-scale networks as the task unfolds. This rapid, state-dependent reorganization cannot be captured with offline approaches, which reflect slower, cumulative aftereffects rather than immediate network adaptations and 4) associated with the previous point, online perturbation ensures that observed effects are directly tied to the task context and processing demands, rather than reflecting global shifts in excitability that may generalize across tasks or states. Therefore, we think that an online approach as applied in the present study was best suited to address the causal and state-dependent mechanisms of network adaptation. We better explain our motivation in the introduction line 122:

“In this study, we aim to address these challenges by combining an online TMS perturbational approach targeting EVA and MT with fMRI to accurately address the causal and state-dependent mechanisms of perceptual decision-making networks adaptation.”

4. Lines 364 - 368. Please replace "informax" with "infomax". The authors use ICA to identify the networks of interest. They also used PCA to reduce the dimensionality of their data. Can the authors please specify the average number of dimensions to which the data were reduced using PCs, and the average number of ICs they sought? This information is relevant.

Thank you for noticing the typo and for giving us the opportunity to clarify the ICA computations.

Regarding the PCA step, this is an important point indeed because PCA can drastically affect the data structure. We used two reduction steps. We first performed a subject-level principal component analysis (PCA) with the number of principal components as 16 (PC1 = 16) on each subject's fMRI data corresponding to approx. 90% of the explained variance, and group-level PCA with the number of principal components being 8 (PC2 = 8) on the reduced and concatenated data. Subsequently, we performed an ICA with Infomax (Bell and Sejnowski, 1995) on the PCA-reduced data, resulting in 8 group-level components. These numbers of subject-level and group-level components were chosen to improve the later back-reconstruction and to produce refined networks which should match with anatomical and functional segmentation (Erhardt et al., 2011; Wang et al., 2006).

We have added this information line 353:

“We then performed a subject-level principal component analysis (PCA) with the number of principal components as 16 on each subject's fMRI data and group-level PCA with the number of principal components being 8 on the reduced and concatenated data.”

Concerning the average number of ICs, we used Maximum Description Length (MDL) and Akaike's criteria (AIC). MDL prioritizes the most compressed data description, while AIC selects the model that best explains the data with the fewest parameters, with lower scores indicating a better fit. Both methods are used for model selection, determining the optimal number of ICs in our case. The number of extracted ICs (8) can appear lower than what is typically reported in resting-state fMRI studies. However, several features of the present dataset explain this choice. First, the concurrent online TMS-fMRI paradigm induces strong, stereotyped, and time-locked activity patterns, which dominate the variance structure of the data and reduce the relative contribution of weaker or more distributed sources. As a result, the effective dimensionality of the dataset is lower compared to unconstrained resting-state acquisitions. Second, our focus was on capturing the large-scale functional networks directly engaged by TMS perturbation during motion processing. Increasing the dimensionality might have risked fragmenting these networks into smaller subcomponents without adding interpretative value. The results of the procedure thus represent a balance between capturing the major, reproducible TMS-evoked networks while avoiding overfitting to noise.

This information is mentioned in the Method section line 366:

"Maximum Description Length (MDL) and Akaike's criteria were applied to estimate the number of ICs in our data. Using principal component analysis, individual data was reduced. Then, the infomax algorithm (Bell and Sejnowski, 1995) was applied on the PCA-reduced data for the group ICA and estimated 8 components"

We have added a short explanation right after:

"..., probably explained by the nature of online TMS-fMRI data which induces strong and stereotyped activity patterns that dominate the variance structure of the data, resulting in a lower effective dimensionality compared to resting-state acquisitions"

5. Section 3.3, lines 505 - 529. The authors used the IC number to indicate the networks of interest. See, for example, lines 507-509, "Five networks (IC1, IC3, IC4, IC7, and IC8) were labelled as TMS-related networks because they overlapped with the stimulation site (Figure 4A and Supplementary Table S5 for the MNI coordinates of the IC)." Then the reader has to wait until the supplementary material to see which networks the ICs represent. I suggest moving those tables to the main text or including more information about the networks in that section. Fig. 4 also should contain that information.

The surface-representation of the networks are presented in Figure 4. As suggested, the previous supplementary table S5, which depicts the cluster region, extent and the MNI coordinates of the peak activity has been integrated into the main text as Table 3.

6. Overall, the literature review should be deepened, and more complete original papers should be added to the main manuscript where needed. See, for instance, <https://doi.org/10.1038/s41596-025-01182-4>.

We thank the reviewer for the suggestion. This consensus guideline paper has been added together with a couple of recent or relevant papers on TMS-fMRI coupling and visual processing. We kindly refer the reviewer to the additional reference list at the end of this document.

7. Please add to Figure caption 1.A. that the coil was placed in the right EVA and right MT.

We thank the reviewer for this comment. The information has been added to the caption.

Also, check the spelling in line 232. I think you meant "coil" instead of "oil".

We actually used capsule with vegetable oil inside to visualize the coil's orientation on the MRI images, especially to be able to readjust the coil position before starting the experiment.

I hope the authors find my comments helpful to improve the impact of the manuscript.

We thank you once more for the constructive comments and overall appreciation of our work.

Referee #3:

This is a well-designed and timely study that has the potential to advance our understanding of how transient perturbation of different visual areas by TMS influences local BOLD activity, network organization, and motion perception behavior. The combination of TMS bursts well timed with ongoing task processing with concurrent fMRI as readout allows insights into what the authors call "causal disconnectomics." Its really hard to interpret the findings since even though we have a causal tool at hand, the BOLD changes could be interpreted in a "virtual lesion" framework, as compensatory effects, hyperactivation etc... The authors make a good job in keeping interpretation balances. The manuscript is clearly written and represents a strong contribution to the field. I offer the following comments for improvement:

We sincerely appreciate Reviewer #2's positive assessment of our work. Below, we provide detailed responses to each of the comments.

Major:

1. Figures - color bars are missing: A major methodological concern is that the statistical brain map figures lack color bars. Without color bars, readers cannot interpret the magnitude or statistical thresholds of the activations, and the images remain only illustrative. Properly labeled color bars (e.g., mapping colors to t-values or p-values) should be included in all statistical figures to make the data interpretable and reproducible.

We apologize for the omission. We have added the color-coded z scores bar in all the fMRI figures.

Minor:

2. Stimulation timing and state-dependency: The chronometric design is excellent, and the early vs late TMS effects are a strength. Results highlight that stimulation timing relative to neural processing stages is crucial. This ties directly into the principle of state-dependency-TMS interacts with ongoing neural activity and task engagement, not simply exerting uniform "excitatory" or "inhibitory" effects. It would be valuable to elaborate this point more explicitly in the Discussion. For context, recent chronometric TMS-fMRI studies (e.g., Grosshagauer et al., 2024) have shown that downstream BOLD responses differ depending on whether stimulation is delivered during different phases in task and at rest. This comparison with other literature could make state-dependent TMS effects more integrated.

We thank the reviewer for this insightful suggestion. We agree that our findings fit very well within the framework of state-dependent TMS effects. We have now expanded the Discussion to emphasize that the observed differences between early and late stimulation are consistent with the idea that TMS interacts with ongoing neural dynamics rather than exerting uniform effects. This further highlights the value of chronometric perturbation designs for disentangling the causal contribution of targeted regions within distributed networks. We have added line 633:

“Beyond the active/rest comparison, we showed that TMS interacts with the ongoing neural context, such that stimulation timing relative to task processing stages determines both behavioral consequences and downstream network responses. Early stimulation of EVA and MT selectively impaired direction discrimination and induced distinct patterns of network reorganization, whereas later stimulation had no effect for EVA. This temporal specificity is in line with recent chronometric TMS-fMRI studies (Grosshagauer et al., 2024). Together, these findings reinforce the principle that TMS should be understood as a state-dependent perturbation, whose causal impact is dynamically shaped by neural processing and behavioral context.”

3. Control conditions: The inclusion of noTMS and control-site (M1) conditions is a strong aspect of the study. It could still be helpful for the authors to discuss whether sham TMS or control-site stimulation during fMRI might serve as additional baselines in future work, especially to control for potential nonspecific auditory or somatosensory effects. In the current study this would be beyond the scope though.

We appreciate the reviewer’s point. We agree that including additional baselines, such as sham TMS or stimulation of a control site during fMRI, could further help to disentangle nonspecific auditory or somatosensory effects. We have now acknowledged this point in the Discussion and highlighted it as an important direction for future work line 778:

“Another important consideration for future studies is the inclusion of additional control baselines, such as sham TMS or stimulation of a control site during fMRI. These

approaches could provide further leverage to dissociate specific network effects of TMS from nonspecific auditory or somatosensory contributions.”

4. Interpretation of local BOLD responses: The observed local BOLD "over-activation" beneath the coil during motion trials, coupled with behavioral impairment. This pattern is compatible with models of TMS adding "neural noise" that reduces signal-to-noise ratio in task-relevant networks. I suggest the authors expand the Discussion on what positive BOLD increases mean physiologically: whether they reflect excitation, disinhibition, or compensatory activity. Referencing the debate in the literature (e.g., Rafiei & Rahnev, 2022) will strengthen this interpretation.

We thank the reviewer for this important suggestion. We expanded the Discussion section to more explicitly consider physiological mechanisms that could underlie the local positive BOLD increases, observed exclusively during motion trials. More precisely, in the revised manuscript we now summarize competing physiological accounts (i.e., direct excitation, disinhibition, compensatory recruitment and increased neural variability / reduced signal-to-noise ratio), and discuss our results in the context of Rafiei & Rahnev’s review, arguing that TMS effects on local BOLD are variable and task/state dependent. The new text appears line 591:

“The observed local positive BOLD underneath the coil paired with impaired motion performance can be interpreted in several, non-mutually exclusive ways. One possibility is that the positive BOLD reflects net excitation, meaning an overall increase in local synaptic or spiking activity that produces higher metabolic demand and thus increases BOLD signal (Howarth et al., 2020). However, concurrent TMS–fMRI studies revealed a more complex picture: while suprathreshold TMS to primary motor or visual cortex often produces local BOLD increases (likely driven by downstream motor or phosphene elicitation), TMS to many cortical targets does not reliably increase local BOLD at rest, and local BOLD effects appear strongly state-dependent (Rafiei and Rahnev, 2021). In particular, Rafiei & Rahnev (2022) argue that TMS can evoke alternating periods of increased and decreased firing that may cancel in terms of net metabolic signal, and that task engagement can unmask different hemodynamic outcomes (Rafiei and Rahnev, 2022).

An alternative account is that the positive BOLD reflects disinhibition (reduced local inhibitory drive) or a shift in excitation–inhibition balance: disinhibition can produce increased local firing and hemodynamic signal while simultaneously disrupting precise, information-bearing spiking patterns required for task performance. One can also speculate that this over-activity reflects a compensatory recruitment. TMS may perturb the targeted area and cause increased local BOLD because neighboring or connected populations increase activity to compensate for perturbation, yet this compensatory activity is functionally inefficient and does not prevent behavioral impairment.

Finally, this coupling between local BOLD increase and decreased performance suggests that the additional activity may reflect the accumulation of task-irrelevant neural noise (Bancroft et al., 2014; Ruff et al., 2009). Such a mechanism aligns with established models of TMS-induced disruption, where stimulation is thought to activate neuronal populations that are not optimally tuned to the task-relevant feature—in this case, motion direction (Silvanto and Pascual-Leone, 2008). As a result, the effective signal-to-noise ratio within the motion-sensitive network would be reduced, impairing accurate motion discrimination. While we cannot firmly decide for one mechanism over the

others, these results reinforce the idea that local BOLD changes after TMS can reflect mixed effects rather than a simple monotonic “excitation” signal. Future work combining concurrent TMS with measures that index the excitation-inhibition balance (e.g., MR spectroscopy, laminar fMRI, functional DWI or invasive recordings) would help dissociate these possibilities.”

5. Network-level analysis and causal disconnectomics: The ICA and graph-theoretical results are compelling, showing distinct network reorganizations after EVA vs MT stimulation. The finding that clustering coefficient predicts behavioral impairment following MT stimulation is especially novel and should be emphasized. This link between topology and behavioral vulnerability is an important contribution to causal disconnectomics. It would also be valuable to discuss why EVA's network configuration might be more resilient (higher small-worldness, local clustering) compared to MT's more distributed pattern.

This is indeed one of the most interesting findings of our study. We now discuss further these differences in network topologies in relationship with behavior in the Discussion line 718:

“Interestingly, in many domains of network science, systems with higher local clustering and small-world properties are more resilient to perturbation because they enable redundant, short-range communication paths (e.g., Eom, 2018). In the brain, similar principles have been observed in studies of lesion impact, epilepsy, and aging, where disruption of highly distributed hubs often has stronger behavioral consequences than perturbation of more locally clustered regions (Hwang et al., 2021; Reber et al., 2021).”

And line 739:

“More globally, these results support the notion that topology-based markers can predict which nodes are “fragile” versus “robust” under external disruption, strengthening the contribution of this work to the emerging framework of causal disconnectomics.”

6. Broader implications: The study nicely connects experimental perturbations to concepts of network resilience and compensation. Another suggestion would be to highlight implications beyond vision—for example, whether similar principles may apply in clinical contexts such as stroke or depression, where causal mapping of network vulnerabilities could guide interventions.

We agree with the reviewer that this methodology could be applied to many clinical contexts, to predict for instance the resilience of functional networks in response to progressive neurodegeneration or focal lesions, and potentially to guide the development of more targeted therapeutic strategies. However, as suggested by Reviewer #2, we chose to remain factual at this stage and have focused the scope of our paper to the exploration of neural network dynamics. However, we kept an opening paragraph in which we mention the potential of the technic line 794:

“Complementing classical lesion studies (Das et al., 2012), our results highlight the utility of TMS-fMRI coupling as a powerful tool for investigating ‘causal disconnectomics’—the causal relationships between localized disruptions and large-scale neural and

behavioural consequences (Egger et al., 2021). This approach holds promise for precisely mapping how local perturbations propagate through brain networks, especially when assessed during active cognitive processing. Ultimately, such techniques may advance our understanding of brain resilience, compensation, and vulnerability in the face of focal damage.”

Additional references

Bancroft, T.D., Hogeveen, J., Hockley, W.E., Servos, P., 2014. TMS-induced neural noise in sensory cortex interferes with short-term memory storage in prefrontal cortex. *Front Comput Neurosci* 8. <https://doi.org/10.3389/fncom.2014.00023>

Bell, A.J., Sejnowski, T.J., 1995. An information-maximization approach to blind separation and blind deconvolution. *Neural Comput* 7, 1129–1159. <https://doi.org/10.1162/neco.1995.7.6.1129>

Blanco, R., Preti, M.G., Koba, C., Ville, D.V.D., Crimi, A., 2024. Comparing structure–function relationships in brain networks using EEG and fNIRS. *Sci Rep* 14, 28976. <https://doi.org/10.1038/s41598-024-79817-x>

Das, A., Tadin, D., Huxlin, K.R., 2014. Beyond Blindsight: Properties of Visual Relearning in Cortically Blind Fields. *Journal of Neuroscience* 34, 11652–11664. <https://doi.org/10.1523/JNEUROSCI.1076-14.2014>

Farooqui, M., Ortega-Gutierrez, S., Hernandez, K., Torres, V.O., Dajles, A., Zevallos, C.B., Quispe-Orozco, D., Mendez-Ruiz, A., Manzel, K., Ten Eyck, P., Tranel, D., Karandikar, N.J., Ortega, S.B., 2022. Hyperacute immune responses associate with immediate neuropathology and motor dysfunction in large vessel occlusions. *Ann Clin Transl Neurol* 10, 276–291. <https://doi.org/10.1002/acn3.51719>

Glick, C., Gajawelli, N., Sun, Y., Badami, F., Saggari, M., Etkin, A., 2025. Concurrent single-pulse TMS-fMRI dataset to reveal the causal connectome in healthy and patient populations. *Sci Data* 12, 1081. <https://doi.org/10.1038/s41597-025-05377-y>

Glick, C., Gajawelli, N., Sun, Y., Badami, F., Saggari, M., Etkin, A., 2024. Concurrent single-pulse (sp) TMS/fMRI to reveal the causal connectome in healthy and patient populations. *bioRxiv* 2024.09.25.614833. <https://doi.org/10.1101/2024.09.25.614833>

Grosshagauer, S., Woletz, M., Vasileiadi, M., Linhardt, D., Nohava, L., Schuler, A.-L., Windischberger, C., Williams, N., Tik, M., 2024. Chronometric TMS-fMRI of personalized left dorsolateral prefrontal target reveals state-dependency of subgenual anterior cingulate cortex effects. *Mol Psychiatry* 29, 2678–2688. <https://doi.org/10.1038/s41380-024-02535-3>

He, H., Sun, X., Doose, J., Faller, J., McIntosh, J.R., Saber, G.T., Huffman, S., Hong, L., Pantazatos, S.P., Yuan, H., McTeague, L.M., Goldman, R.I., Brown, T.R., George, M.S., Sajda, P., 2024. TMS-induced modulation of brain networks and its associations to rTMS treatment for depression: a concurrent fMRI-EEG-TMS study. <https://doi.org/10.1101/2024.12.24.24319609>

Howarth, C., Mishra, A., Hall, C.N., 2020. More than just summed neuronal activity: how multiple cell types shape the BOLD response. *Philosophical Transactions of the Royal Society B: Biological Sciences* 376, 20190630. <https://doi.org/10.1098/rstb.2019.0630>

Jackson, J.B., Feredoes, E., Rich, A.N., Lindner, M., Woolgar, A., 2021. Concurrent neuroimaging and neurostimulation reveals a causal role for dlPFC in coding of task-relevant information. *Commun Biol* 4, 588. <https://doi.org/10.1038/s42003-021-02109-x>

Lai, T.W., Zhang, S., Wang, Y.T., 2014. Excitotoxicity and stroke: Identifying novel targets for neuroprotection. *Progress in Neurobiology*, 2013 Pangu Meeting on Neurobiology of Stroke and CNS Injury: Progresses and Perspectives of Future 115, 157–188. <https://doi.org/10.1016/j.pneurobio.2013.11.006>

Lim, S., Radicchi, F., van den Heuvel, M.P., Sporns, O., 2019. Discordant attributes of structural and functional brain connectivity in a two-layer multiplex network. *Sci Rep* 9, 2885. <https://doi.org/10.1038/s41598-019-39243-w>

Litwińczuk, M.C., Muhlert, N., Cloutman, L., Trujillo-Barreto, N., Woollams, A., 2022. Combination of structural and functional connectivity explains unique variation in specific domains of cognitive function. *NeuroImage* 262, 119531. <https://doi.org/10.1016/j.neuroimage.2022.119531>

Ma, S., Huang, T., Qu, Y., Chen, X., Zhang, Y., Zhen, Z., 2022. An fMRI dataset for whole-body somatotopic mapping in humans. *Sci Data* 9, 515. <https://doi.org/10.1038/s41597-022-01644-4>

Rafiei, F., Rahnev, D., 2022. TMS Does Not Increase BOLD Activity at the Site of Stimulation: A Review of All Concurrent TMS-fMRI Studies. *eNeuro* 9. <https://doi.org/10.1523/ENEURO.0163-22.2022>

Rafiei, F., Rahnev, D., 2021. Does TMS increase BOLD activity at the site of stimulation?

Ruff, C.C., Driver, J., Bestmann, S., 2009. Combining TMS and fMRI. *Cortex* 45, 1043–1049. <https://doi.org/10.1016/j.cortex.2008.10.012>

Schuler, A.-L., Hartwigsen, G., 2025. The potential of interleaved TMS-fMRI for linking stimulation-induced changes in task-related activity with behavioral modulations. *Brain Stimulation* 18, 37–51. <https://doi.org/10.1016/j.brs.2024.12.1190>

Silvanto, J., Pascual-Leone, A., 2008. State-Dependency of Transcranial Magnetic Stimulation. *Brain Topogr* 21, 1–10. <https://doi.org/10.1007/s10548-008-0067-0>

Woolgar, A., Feredoes, E., Assem, M., Bassil, Y., Bergmann, T.O., Beynel, L., Burke, M., Cash, R.F.H., Comeau, R.M., Correia, M.M., Genc, E., Hartwigsen, G., Jackson, J.B., Kienle, M., Kunz, P., Leticevscaia, O., Luber, B., Lueckel, M., Mathiesen, C., Michael, E., Numssen, O., Oathes, D.J., Rosen, A.C., Schuhmann, T., Schuler, A.-L., Scrivener, C.L., Thielscher, A., Tik, M., Todorov, Y., Vasileiadi, M., Windischberger, C., Hermiller, M.S., Sack, A.T., 2025. Consensus guidelines for the use of concurrent TMS-fMRI in cognitive and clinical neuroscience. *Nat Protoc* 1–17. <https://doi.org/10.1038/s41596-025-01182-4>

Dear Dr Raffin,

Re: JP-RP-2025-289699R1 "**Causal disconnectomics of motion perception networks: insights from TMS-induced BOLD responses**" by Estelle Raffin, Roberto Felipe Salamanca-Giron, Krystel R Huxlin, Olivier Reynaud, Loan Mattera, Roberto Martuzzi, and Friedhelm Hummel

Thank you for submitting your manuscript to The Journal of Physiology. It has been assessed by a Reviewing Editor and by 2 expert referees and we are pleased to tell you that it is acceptable for publication following satisfactory revision.

REVISION CHECKLIST:

Please upload two versions of your manuscript text: one with all relevant changes highlighted and one clean version with no changes tracked. The manuscript file should include all tables and figure legends, but each figure/graph should be uploaded as separate, high-resolution files. The journal is now integrated with Wiley's Image Checking service. For further details, see: <https://www.wiley.com/en-us/network/publishing/research-publishing/trending-stories/upholding-image-integrity-wileys->

image-screening-service

We look forward to receiving your revised submission.

Yours sincerely,

Richard Carson
Senior Editor
The Journal of Physiology

EDITOR COMMENTS

Reviewing Editor:

Thank you for submitting your revision to The Journal of Physiology. It has been reviewed again by the two referees, who do not have any concerns anymore. The only open point is the requirement for publicly available code. Please provide code for your analyses in a repository or specify why this is not possible. Congratulations on your manuscript.

REFEREE COMMENTS

Referee #2:

The authors have addressed all my comments, and therefore, I recommend the publication of this manuscript in its present form.

Referee #3:

The authors have addressed my comments. I congratulate to this important paper and suggest prompt production.

END OF COMMENTS

Reviewing Editor:

Thank you for submitting your revision to The Journal of Physiology. It has been reviewed again by the two referees, who do not have any concerns anymore. The only open point is the requirement for publicly available code. Please provide code for your analyses in a repository or specify why this is not possible. Congratulations on your manuscript.

Thank you very much. As requested, we have uploaded our codes on a public repository here: DOI [10.5281/zenodo.17511167](https://doi.org/10.5281/zenodo.17511167). This is now mentioned in the text, in the Data availability statement section.

Dear Dr Raffin,

Re: JP-RP-2025-289699R2 "**Causal disconnectomics of motion perception networks: insights from TMS-induced BOLD responses**" by Estelle Raffin, Roberto Felipe Salamanca-Giron, Krystel R Huxlin, Olivier Reynaud, Loan Mattered, Roberto Martuzzi, and Friedhelm Hummel

We are pleased to tell you that your paper has been accepted for publication in The Journal of Physiology.

Yours sincerely,

Richard Carson
Senior Editor
The Journal of Physiology

IMPORTANT POINTS TO NOTE FOLLOWING ACCEPTANCE OF YOUR PAPER:

- You can help your research get the attention it deserves! Check out Wiley's free Promotion Guide for best-practice recommendations for promoting your work at: www.wileyauthors.com/eeo/guide. You can learn more about Wiley Editing Services which offers professional video, design, and writing services to create shareable video abstracts, infographics, conference posters, lay summaries, and research news stories for your research at: www.wileyauthors.com/eeo/promotion.

- If you would like to receive our 'Research Roundup', a monthly newsletter highlighting the cutting-edge research published in The Physiological Society's family of journals (The Journal of Physiology, Experimental Physiology, Physiological Reports, The Journal of Nutritional Physiology and The Journal of Precision Medicine: Health and Disease), please click this link, fill in your name and email address and select 'Research Roundup': <https://www.physoc.org/journals-and-media/membernews>